# LangPrecip: Language-Aware Multimodal Precipitation Nowcasting

Xudong Ling [* 1]   Chaorong Li [* 2]   Tianxi Huang [3]   Qian Dong [1]   Guiduo Duan [1]

## Abstract

Short-term precipitation nowcasting is inherently under-constrained due to limited historical observation windows: identical observations can lead to multiple plausible future trajectories, especially for extreme events. Existing generative methods rely solely on visual features and lack explicit constraints on precipitation motion semantics, resulting in ambiguous dynamics, blurred details, and unstable predictions. We propose LangPrecip, the first language-guided precipitation nowcasting framework, and contribute LangPrecip-160K, a large-scale radar-text paired dataset with 160K annotated sequences. LangPrecip addresses the under-constrained challenge by leveraging natural-language motion descriptions as explicit semantic constraints to reduce motion ambiguity and introducing a dual-path wavelet consistency unfolding decoder that enforces physical data fidelity during latent-to-pixel reconstruction. By reformulating nowcasting as semantically constrained trajectory generation under the Rectified Flow paradigm with model-based decoder optimization, LangPrecip produces sharper and more physically consistent forecasts. Experiments on Swedish and MRMS benchmarks demonstrate substantial improvements over state-of-the-art vision-only methods, achieving over 60% and 19% relative gains in heavy-rainfall CSI at 80-minute lead time with enhanced spatial detail preservation. The dataset and code are available at github.com/UESTC-LXD/LangPrecip.

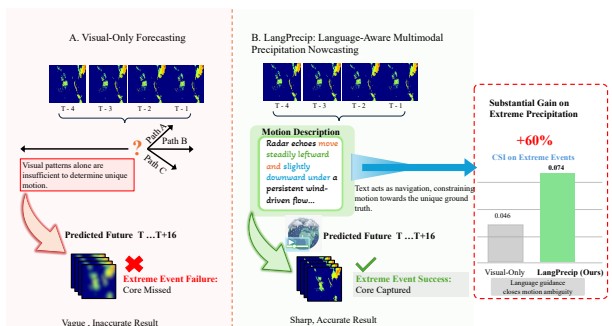

*Figure 1.* Visual-only nowcasting suffers from motion ambiguity under limited radar history. LangPrecip resolves this ambiguity by using language-based motion descriptions to constrain precipitation evolution and improve extreme event prediction.

## 1. Introduction

Accurate precipitation nowcasting is a crucial component of disaster prevention and early warning systems (Ravuri et al., 2021). Precipitation nowcasting has evolved from numerical weather prediction and optical-flow-based extrapolation to data-driven regression and recent probabilistic generative models. While advances in diffusion models (Ho et al., 2020) and Rectified Flow (Liu et al., 2022; Lipman et al., 2022) have improved uncertainty representation and enabled diverse predictions, reliably resolving the evolution of extreme precipitation under limited observations remains an open challenge.

**Limitations of Vision-Only Nowcasting.** Most precipitation nowcasting methods adopt a vision-only paradigm, relying solely on short radar or satellite sequences. As illustrated in Figure 1A, limited historical observations (typically 4–8 frames) are insufficient to uniquely determine physically consistent motion trajectories (Yan et al., 2024; Pulkkinen et al., 2019). Classical deterministic models (ConvLSTM (Shi et al., 2015), Transformers (Gao et al., 2022; Wang et al., 2025b)) collapse multiple plausible futures into over-smoothed forecasts that fail to capture extreme precipitation (Ravuri et al., 2021; Cambier van Nooten et al., 2023). Recent generative approaches (GANs (Ravuri et al., 2021), diffusion models (Yu et al., 2024; Gao et al., 2023; Ling et al., 2024b; Feng et al., 2025)) introduce stochastic sampling for diversity, yet without explicit motion semantic constraints, they often produce visually plausible but physi-

*Equal contribution [1]Laboratory of Intelligent Collaborative Computing, University of Electronic Science and Technology of China (UESTC), Chengdu 611731, China [2]School of Computer Science and Technology (School of Artificial Intelligence), Yibin University, Yibin 644000, China [3]College of Humanities and General Education, Chengdu Textile College, Chengdu 611731, China. Correspondence to: Guiduo Duan <duanguiduo@163.com>.

*Proceedings of the 43rd International Conference on Machine Learning*, Seoul, South Korea. PMLR 306, 2026. Copyright 2026 by the author(s).

cally inconsistent evolutions (Lin et al., 2025; Tu et al., 2025; Gong et al., 2024; Kurki et al., 2025). In addition to the uncertainty of future dynamics, latent generative nowcasting (Ling et al., 2024b; Li et al., 2024) introduces additional challenges during the reconstruction phase. The aggressive spatiotemporal compression in latent-space modeling makes the decoding process from latent space to radar a severely ill-posed inverse problem, often resulting in blurred spatial details and temporal inconsistencies, even when the predicted motion dynamics are reasonable. Identical radar sequences can correspond to multiple valid futures—translation, splitting, intensification, or dissipation (Ravuri et al., 2021; Veillette et al., 2020). While stochastic generation samples across possibilities, it cannot enforce trajectory consistency, degrading intensity-sensitive metrics (high-threshold CSI, fine-scale FSS) (Cambier van Nooten et al., 2023; Yan et al., 2024). This indicates that visual stochasticity alone is insufficient, motivating explicit semantic guidance beyond pixel observations.

**Inspiration from Text-Guided Generation.** Recent text-to-video generation methods have demonstrated that linguistic descriptions can effectively guide complex spatiotemporal modeling. Works such as Imagen Video (Ho et al., 2022), Make-A-Video (Singer et al., 2022), Video LDM (Blattmann et al., 2023), and more recent Transformer-based architectures (Ma et al., 2025; Peng et al., 2025; Yang et al., 2024; Wan et al., 2025) show that text provides high-level semantic priors via cross-modal attention, effectively constraining motion patterns and reducing ambiguity in scenarios with diverse evolution paths. This success raises a natural question: can similar language guidance resolve the inherent ambiguities in precipitation nowcasting?

**Multimodal Precipitation Nowcasting.** Despite growing interest in multimodal learning for meteorology, language-guided precipitation nowcasting remains largely unexplored. Existing climate foundation models (Nguyen et al., 2023; Bodnar et al., 2025) operate at coarse spatiotemporal resolutions unsuitable for fine-grained radar-based nowcasting, while multimodal benchmarks such as Chaos-Bench (Nathaniel et al., 2024) focus on continuous atmospheric variables rather than discrete precipitation structures. Recent vision-language efforts in meteorology (Chen et al., 2024) restrict language to static semantic tasks and do not constrain dynamic motion generation. In contrast, large-scale vision datasets (Bain et al., 2021; Xue et al., 2022) demonstrate that language can encode motion-level semantics beyond pixel observations—a capability entirely absent in radar nowcasting, where datasets lack semantic annotations. As a result, forecasting from short radar histories remains fundamentally under-constrained, with multiple physically plausible futures that vision-only methods fail to disambiguate.

To address this, we propose **LangPrecip**, a language-guided forecasting framework that introduces natural-language motion descriptions as explicit semantic constraints to regularize under-determined radar dynamics (Figure 1). Our main contributions are:

- We propose **LangPrecip**, a language-aware multimodal precipitation nowcasting framework that reformulates forecasting as a dynamically conditioned generation problem under the Rectified Flow framework. High-level motion semantics extracted from language condition the learning of latent radar dynamics, providing stable motion priors for highly under-constrained short-window forecasts. This language-aware conditioning reduces motion collapse and trajectory instability in vision-only models, leading to improved accuracy in extreme precipitation prediction.

- We design a **Wavelet Consistency Unfolding Decoder (WCUD)** integrating learned upsampling with wavelet consistency unfolding—an iterative optimization that explicitly enforces observation fidelity and multi-scale sparsity through unfolded proximal operators. Tailored to precipitation fields, this decoder substantially improves reconstruction sharpness and temporal coherence over standard VAE decoders.

- We curate **LangPrecip-160K**, the first radar-language dataset where natural-language motion descriptions encode physically meaningful priors (advection direction, growth/decay patterns, and structural evolution), enabling explicit disambiguation of plausible futures beyond pixel-level observations alone.

## 2. Method

### 2.1. LangPrecip-160K: A Multimodal Precipitation Nowcasting Dataset

We construct LangPrecip-160K (Figure 2), a large-scale dataset comprising 100K events from Swedish radar data and 60K from MRMS, all extracted via a unified sliding-window strategy. Each event spans 100 minutes (20 frames at 5-minute intervals), using the first 4 frames as input to predict the subsequent 16. We apply consistent preprocessing across both sources, including spatial cropping, quality filtering for data integrity, and temporal alignment. Evaluation follows established nowcasting benchmarks (Ling et al., 2024b; Ravuri et al., 2021) with precipitation thresholds of 0.06 mm/h (corresponding to the radar detection limit of 5 dBZ) and 6.3 mm/h for the Swedish dataset, and 1 and 8 mm/h for MRMS.

*Figure 2.* LangPrecip architecture diagram

### 2.1.1. MOTION DESCRIPTION ANNOTATION

Motion descriptions provide qualitative semantic constraints on precipitation dynamics using natural language (e.g., "echoes move leftward and intensify"). This approach enables high inter-annotator consistency, aligns with vision-language model capabilities, and provides sufficient high-level guidance without requiring precise numerical measurements.

**Annotation Process.** We generated motion descriptions using Qwen-VL with meteorology-specific prompts. Vision-language models (VLMs) excel at qualitative visual description, making them well-suited for capturing directional motion and structural evolution without numerical precision. All generated descriptions were reviewed by trained meteorologists for correctness, validating that they reliably encode meteorological motion patterns for model training.

**Language as High-Level Prior.** Motion descriptions extracted from observed frames ($T = 0$–$4$) provide coarse semantic constraints (e.g., prevailing advection direction) that regularize long-horizon forecasts. The language condition biases the generative model toward physically plausible motion regimes while preserving uncertainty in fine-scale evolution (intensity fluctuations, localized deformation), which is learned implicitly from visual observations. This design allows the model to leverage linguistic priors when motion semantics are well-defined, while gracefully degrading to vision-only forecasting when language guidance is weak or absent.

### 2.2. Language-Aware Motion Prediction Modeling

We formulate short-term precipitation nowcasting as a *semantically constrained trajectory generation* problem in latent space. A key challenge is that short historical windows (e.g., 4 frames) provide insufficient constraints, causing trajectories to diverge from physical reality, especially under rapidly evolving weather conditions. To address this, we introduce meteorological language as explicit motion constraints that compensate for limited observational context. We adopt the Rectified Flow framework (Lipman et al.,

2022), which models trajectory dynamics via a deterministic velocity field that transports samples from a prior to the data distribution. We operate on latent representations encoded by a VAE, where semantic motion information is incorporated as conditioning to constrain trajectory evolution alongside historical radar context. This allows linguistic semantics to directly guide the generative dynamics, preventing trajectory deviation while preserving uncertainty modeling.

### 2.2.1. TRAINING OBJECTIVE

We train a Rectified Flow model (Liu et al., 2022) to learn semantically constrained dynamics in latent space. A 2D VAE encodes radar sequences from $256 \times 256$ to $32 \times 32$ latents (downsampling factor $8$). The flow model predicts future latents ($T = 16$) conditioned on historical context ($T = 4$) and motion descriptions.

Given target latent $x_1$ (Latent Z) and noise $x_0 \sim \mathcal{N}(0, I)$, we define the interpolation $x_t = tx_1 + (1 - t)x_0$ and minimize the velocity matching loss:

$$\mathcal{L} = \mathbb{E}_{x_0, x_1, c_{\text{ctx}}, t} \left\| u(x_t, c_{\text{ctx}}, t; \theta) - (x_1 - x_0) \right\|_2^2, \quad (1)$$

where $u(\cdot)$ is conditioned on $c_{\text{ctx}} = \{X_{0:4}, m\}$: encoded historical frames and motion description $m$ via T5-XXL (Chung et al., 2024). Crucially, $m$ is generated exclusively from the first four observed frames to ensure causal consistency, while the model is supervised on full 20-frame trajectories to learn long-range dynamics. This design compensates for insufficient observational constraints through linguistic motion priors.

### 2.3. Wavelet Consistency Unfolding Decoder

Standard VAE decoders often produce blurry and temporally inconsistent radar reconstructions, as they learn a direct latent-to-pixel mapping without explicitly modeling the physical observation process. In precipitation nowcasting, latent features undergo aggressive spatiotemporal compression (e.g., $8\times$ downsampling), inevitably discarding fine-grained spatial and temporal details during encoding. Due to this severe information loss, standard neural decoders cannot

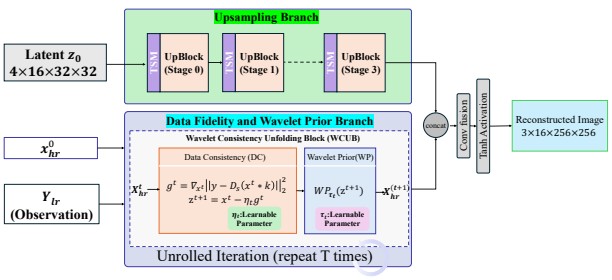

$x_{hr}^0$: Initial HR state (Upsample from LR observation)     $Y_{lr}$: Low-resolution observation (used in DC)

*Figure 3.* Wavelet Consistency Unfolding Decoder architecture with Upsampling Branch (top) and Data Fidelity and Wavelet Prior Branch (bottom) for iterative observation-consistent refinement.

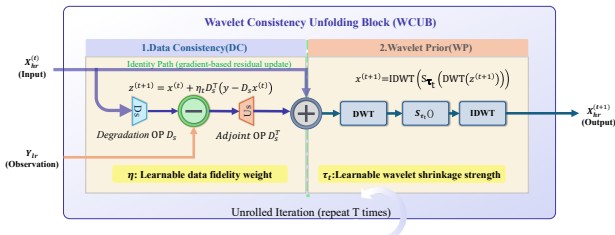

*Figure 4.* Wavelet Consistency Unfolding Block. The Data Consistency (DC) module enforces fidelity to latent observations via back-projection, while the Wavelet Prior (WP) module restores fine details through wavelet-domain shrinkage.

recover missing high-frequency information and typically produce over-smoothed outputs that lack sharp precipitation boundaries.

To address this issue, we propose a Wavelet Consistency Unfolding Decoder (WCUD) that integrates model-based optimization with deep feature learning. Specifically, an **Upsampling Branch** recovers global spatiotemporal structure using UpBlocks with Temporal Shift Modules (TSM) (Lin et al., 2019) for cross-frame alignment, while a **Data Fidelity and Wavelet Prior Branch** explicitly enforces observation consistency and restores fine-scale details through iterative refinement (Figure 3).

### 2.3.1. DATA FIDELITY AND WAVELET PRIOR BRANCH

We explicitly model the latent-to-observation degradation process and formulate reconstruction as an optimization problem. Assume the latent observation $y \in \mathbb{R}^{H_l \times W_l}$ is generated from a high-resolution radar field $x \in \mathbb{R}^{H \times W}$ via

$$y = D_s(x * k) + n, \qquad (2)$$

where $D_s(\cdot)$ denotes downsampling by factor $s = 8$, $k$ is a blur kernel, and $n$ represents encoding noise. Direct inversion of this process is unstable due to severe information loss and noise amplification.

To regularize this reconstruction, we impose a wavelet spar-

sity prior and solve the following optimization problem:

$$\hat{x} = \arg\min_x \underbrace{\|y - D_s(x * k)\|_2^2}_{\text{data fidelity}} + \lambda \underbrace{R(x)}_{\text{wavelet prior}}, \qquad (3)$$

where $R(x) = \|W(x)\|_1$ with $W(\cdot)$ the 2D discrete wavelet transform. This $\ell_1$ wavelet prior is well-suited to radar fields, which exhibit piecewise-smooth regions with sharp precipitation boundaries, and admits a closed-form proximal operator enabling stable optimization unfolding.

Instead of solving this problem iteratively at inference time, we unfold the optimization into $T$ learnable *Wavelet Consistency Unfolding Blocks* ($T = 4$ in our implementation):

$$x^0 = U_s(y), \qquad x^{t+1} = \mathcal{F}_{\theta_t}(x^t, y), \quad t = 0, \ldots, T-1, \qquad (4)$$

where $U_s$ denotes bilinear upsampling and each block $\mathcal{F}_{\theta_t}$ consists of two complementary modules:

**(1) Data Consistency (DC) Module:** The DC module performs one gradient descent step on the data fidelity term:

$$z^{(t+1)} = x^{(t)} - \eta_t \nabla_{x^{(t)}} \|y - D_s(x^{(t)} * k)\|_2^2. \qquad (5)$$

Computing the gradient yields

$$z^{(t+1)} = x^{(t)} + 2\eta_t \cdot D_s^\top \big(y - D_s(x^{(t)} * k)\big) *^\top k, \qquad (6)$$

where $D_s^\top$ and $*^\top k$ denote the adjoint operators. In practice, the transposed convolution is absorbed into $D_s^\top$, leading to

$$z^{(t+1)} = x^{(t)} + 2\eta_t \cdot D_s^\top \big(y - D_s(x^{(t)} * k)\big). \qquad (7)$$

This back-projection step enforces observation fidelity by correcting reconstruction errors in latent space, but may amplify high-frequency artifacts present in the latent representation.

**(2) Wavelet Prior (WP) Module:** To stabilize the reconstruction and recover sharp structures, we apply the proximal operator of the wavelet $\ell_1$ regularization. Specifically, we decompose $z^{(t+1)}$ into wavelet subbands

$$(L, H_h, H_v, H_d) = W(z^{(t+1)}), \qquad (8)$$

where $L$ denotes the low-frequency approximation and $(H_h, H_v, H_d)$ correspond to horizontal, vertical, and diagonal high-frequency details. Soft thresholding is applied to high-frequency subbands:

$$\tilde{H}_i = \text{sign}(H_i) \cdot \max\{|H_i| - \tau_t, 0\}, \quad i \in \{h, v, d\}, \qquad (9)$$

with learnable threshold $\tau_t$, followed by inverse wavelet reconstruction:

$$x^{(t+1)} = W^\top(L, \tilde{H}_h, \tilde{H}_v, \tilde{H}_d). \qquad (10)$$

This operation suppresses noise from the DC module while retaining salient precipitation features (Figure 4). The two branches are fused to balance spatiotemporal modeling with observation fidelity, yielding sharp and coherent reconstructions.

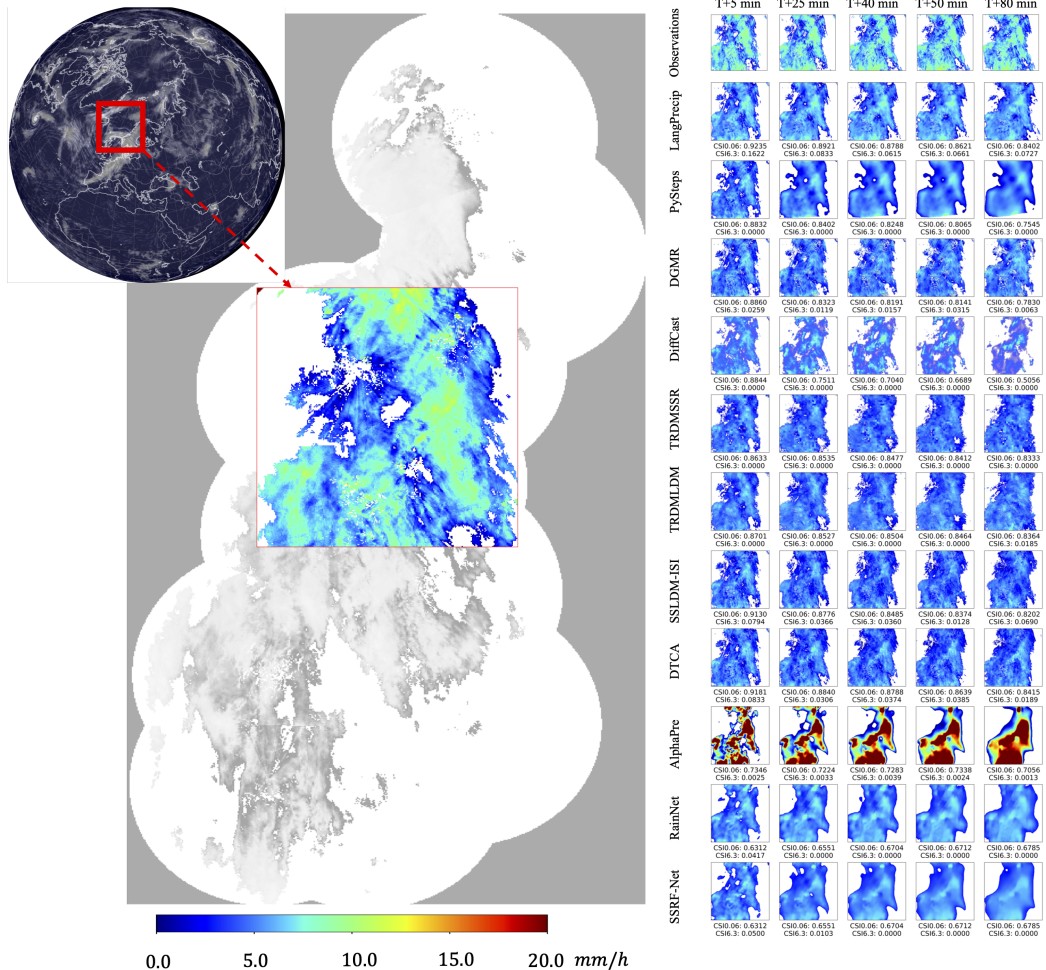

*Figure 5.* Spatial precipitation forecasts at multiple lead times (5–80 minutes) for different models with corresponding skill scores (CSI at 0.06 mm/h and 6.3 mm/h thresholds). Example case from Central Sweden (Östersund region, 63.17°N, 14.63°E) on October 3, 2021, 16:45 UTC.

*Table 1.* The average of all metrics in the Swedish Dataset with a lead time of 80 minutes (16 frames). LangPrecip results use CFG scale = 6.

| Method | CSI $\geq 0.06mm/h \uparrow$ | CSI $\geq 6.3mm/h \uparrow$ | FSS $\uparrow$ | CRPS $\downarrow$ | SSIM $\uparrow$ | LPIPS $\downarrow$ |
|---|---|---|---|---|---|---|
| PySteps(Pulkkinen et al., 2019) | 0.473 | 0.049 | 0.872 | 0.144 | 0.589 | 0.284 |
| DGMR(Ravuri et al., 2021) | 0.433 | 0.039 | 0.882 | 0.143 | 0.702 | 0.204 |
| TRDM-SSR(Ling et al., 2024a) | 0.513 | 0.015 | 0.892 | 0.123 | 0.693 | 0.249 |
| TRDM-LSR(Ling et al., 2024a) | 0.490 | 0.014 | 0.891 | 0.143 | 0.685 | 0.235 |
| DiffCast(Yu et al., 2024) | 0.439 | 0.029 | 0.903 | 0.149 | 0.660 | 0.292 |
| SSLDM-ISI(Ling et al., 2024b) | 0.558 | 0.040 | 0.906 | 0.122 | 0.718 | 0.191 |
| AlphaPre(Lin et al., 2025) | 0.541 | 0.025 | 0.894 | 0.149 | 0.647 | 0.333 |
| DTCA(Li et al., 2024) | 0.572 | 0.046 | 0.911 | **0.115** | **0.732** | **0.180** |
| RainNet(Wang et al., 2025a) | 0.499 | 0.025 | 0.386 | 0.153 | 0.332 | 0.207 |
| SSRF-Net(Luo et al., 2025) | 0.485 | 0.045 | 0.402 | 0.222 | 0.572 | 0.213 |
| LangPrecip(Ours) | **0.586** | **0.074** | **0.923** | 0.132 | **0.732** | 0.193 |

## 3. Results

### 3.1. Experimental Setup

We compare our approach against representative baselines spanning optical-flow-based (PySteps), GAN-based

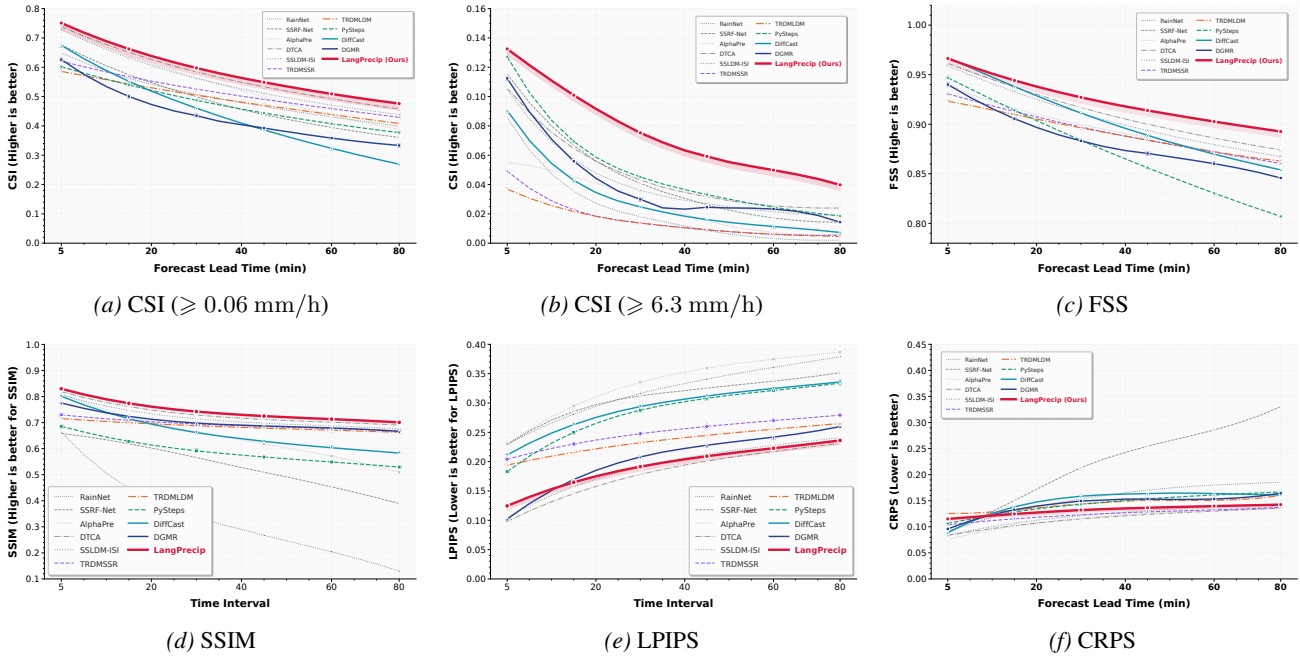

*Figure 6.* Temporal performance evolution on the Swedish dataset across lead times (5–80 minutes). All methods exhibit performance degradation as the prediction horizon increases.

(DGMR, RainNet), diffusion-based (TRDM, DiffCast, SSLDM-ISI), Transformer-based (DTCA), and physics-guided methods (AlphaPre, SSRF-Net).

Evaluation is conducted using both precipitation verification metrics—Critical Success Index (CSI) (Ravuri et al., 2021), Fractions Skill Score (FSS) (Roberts & Lean, 2008), Continuous Ranked Probability Score (CRPS) (Ravuri et al., 2021; Hersbach, 2000)—and perceptual quality metrics—Structural Similarity Index (SSIM) (Wang et al., 2004), Learned Perceptual Image Patch Similarity (LPIPS) (Zhang et al., 2018)—to comprehensively assess forecasting accuracy and visual realism. The complete inference pipeline (including VLM captioning) averages 5.4 s per sample on a single RTX 4090, well within the 5-minute operational radar update cycle.

### 3.2. A case study of precipitation forecasts

Figure 5 compares spatial precipitation forecasts at multiple lead times (5–80 minutes) for a representative Swedish event on October 3, 2021, against the corresponding observations. Predictions from different models are shown from left to right, with skill scores reported under both light (CSI at 0.06 mm/h threshold) and heavy (CSI at 6.3 mm/h threshold) precipitation conditions. As lead time increases, most baselines suffer from structural degradation, including fragmented echoes and displaced high-intensity cores. In contrast, LangPrecip better preserves coherent system structure and propagation, maintaining more accurate lo-

calization of intense precipitation regions at longer lead times, indicating that language-guided motion conditioning effectively stabilizes spatial evolution and improves event detection.

### 3.3. Quantitative Results

Tables 1 and 2 report quantitative results on Swedish and MRMS datasets at 80-minute lead time. LangPrecip yields substantial gains in heavy-precipitation detection, achieving a 60.9% relative CSI improvement on Swedish (from 0.046 to 0.074) and 19.2% on MRMS (from 0.026 to 0.031). In contrast, diffusion-based baselines exhibit severe collapse under high-intensity thresholds (extreme CSI < 0.04), trailing LangPrecip by 50–80% in relative performance despite high SSIM (> 0.66). This gap indicates that visual coherence alone fails to resolve rare-event dynamics. By injecting motion semantics via language priors, LangPrecip regularizes long-range trajectories and mitigates regression to climatological means, substantially improving extreme-event sensitivity.

**Overall Performance:** LangPrecip attains the best FSS (0.923 Swedish, 0.725 MRMS) with competitive CRPS, reflecting superior spatial structure and calibration. Although DTCA achieves lower CRPS through visual attention, it sacrifices physical fidelity, lagging 60.9% behind in extreme CSI on Swedish data—highlighting a misalignment between perceptual optimization and meteorological accuracy.

**Temporal Robustness:** As lead time increases, all methods

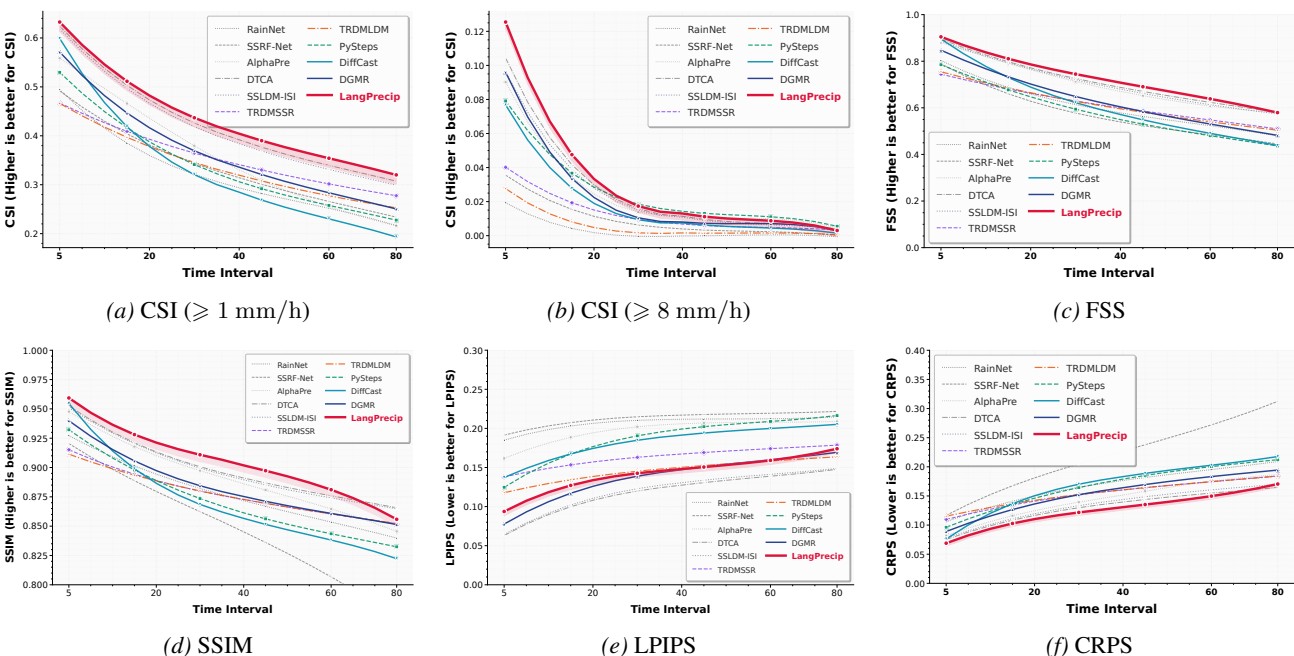

*(a)* CSI ($\geqslant$ 1 mm/h)     *(b)* CSI ($\geqslant$ 8 mm/h)     *(c)* FSS

*(d)* SSIM     *(e)* LPIPS     *(f)* CRPS

*Figure 7.* Temporal performance evolution on the MRMS dataset across lead times (5–80 minutes). All methods exhibit performance degradation as the prediction horizon increases.

*Table 2.* The average of all metrics in the MRMS dataset with a lead time of 80 minutes (16 frames). LangPrecip results use CFG scale = 4.

| Method | CSI $\geq 1mm/h$ ↑ | CSI $\geq 8mm/h$ ↑ | FSS ↑ | CRPS ↓ | SSIM ↑ | LPIPS ↓ |
|---|---|---|---|---|---|---|
| PySteps(Pulkkinen et al., 2019) | 0.338 | 0.024 | 0.579 | 0.167 | 0.871 | 0.187 |
| DGMR(Ravuri et al., 2021) | 0.367 | 0.022 | 0.633 | 0.154 | 0.883 | 0.137 |
| TRDM-SSR(Ling et al., 2024a) | 0.356 | 0.012 | 0.614 | 0.154 | 0.877 | 0.163 |
| TRDM-LSR(Ling et al., 2024a) | 0.338 | 0.014 | 0.613 | 0.155 | 0.877 | 0.146 |
| DiffCast(Yu et al., 2024) | 0.328 | 0.018 | 0.613 | 0.168 | 0.869 | 0.183 |
| SSLDM-ISI(Ling et al., 2024b) | 0.416 | 0.024 | 0.704 | 0.135 | 0.897 | 0.120 |
| AlphaPre(Lin et al., 2025) | 0.360 | 0.024 | 0.699 | 0.144 | 0.891 | 0.197 |
| DTCA(Li et al., 2024) | 0.423 | 0.026 | 0.711 | 0.131 | 0.899 | **0.118** |
| RainNet(Wang et al., 2025a) | 0.320 | 0.002 | 0.606 | 0.153 | 0.877 | 0.207 |
| SSRF-Net(Luo et al., 2025) | 0.336 | 0.009 | 0.573 | 0.219 | 0.844 | 0.213 |
| LangPrecip(Ours) | **0.435** | **0.031** | **0.725** | **0.125** | **0.905** | 0.142 |

degrade, but LangPrecip preserves larger CSI and FSS margins, suggesting that text-encoded temporal context enables more stable long-horizon extrapolation than purely visual motion cues.

### 3.4. Motion Consistency Analysis

Beyond standard quantitative evaluation, we investigate whether the proposed language-aware framework can regulate precipitation dynamics using textual motion descriptions alone, addressing whether high-level linguistic semantics can effectively constrain physical spatiotemporal processes.

In this experiment, all historical radar inputs are removed, and generation is conditioned solely on language describing propagation direction, evolution, and structural changes. This extreme setting isolates the contribution of linguistic semantics and enables a clear assessment of language-guided spatiotemporal control.

To quantify language–dynamics alignment, we compute dense optical flow between consecutive predicted frames and estimate the dominant motion direction as the magnitude-weighted circular mean of flow orientations. This statistic captures bulk advection behavior while remaining robust to local fluctuations, where 0° denotes rightward motion and 90° denotes downward motion.

As shown in Figures 8a and 8b, different textual descrip-

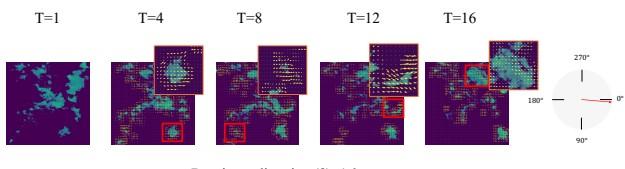

*(a)* Prompt: *The radar echoes surge steadily **rightward with near-perfect directional coherence**, sweeping across the frame as fragmented ... no rupture, no collapse, only a stable, evolving dance of translation, spin, and squeeze, all choreographed by the larger atmospheric current guiding its path.*

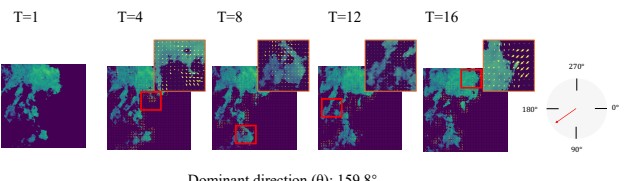

*(b)* Prompt: ***The radar echoes surge leftward and slightly downward in a swift,*** *dragging their fragmented... frame—all motion locked to a single diagonal trajectory, unswerving, as if pulled by an invisible current through the sky.*

*Figure 8.* Dominant motion visualization of text-guided precipitation generation. Motion directions align with textual descriptions; $0°$ and $90°$ denote rightward and downward motion, respectively.

*Table 3.* Effect of language conditioning with CFG on different evaluation metrics across two datasets.

| **Swedish Dataset** (CFG = 6.0) | | | | |
|---|---|---|---|---|
| Prediction settings | CRPS ($\downarrow$) | CSI$\geq$0.06 ($\uparrow$) | CSI$\geq$6.3 ($\uparrow$) | FSS ($\uparrow$) |
| Only radar features | **0.1101** | 0.5668 | 0.0530 | 0.920 |
| Radar features + motion description | 0.1322 | **0.5869** | **0.0746** | **0.923** |
| **MRMS Dataset** (CFG = 4.0) | | | | |
| Prediction settings | CRPS ($\downarrow$) | CSI$\geq$1 ($\uparrow$) | CSI$\geq$8 ($\uparrow$) | FSS ($\uparrow$) |
| Only radar features | 0.1267 | 0.4323 | 0.0302 | 0.722 |
| Radar features + motion description | **0.1256** | **0.4350** | **0.0309** | **0.725** |

tions induce distinctly separable and semantically consistent dominant motion patterns. For instance, a rightward-propagation prompt yields a dominant direction of $1.0°$, whereas a leftward-downward prompt results in $159.8°$, closely reflecting the semantic contrast in the language input.

These results demonstrate that the model achieves semantically guided and physically interpretable motion control. Language-aware constraints promote globally coherent precipitation propagation that complements local pattern modeling, providing a mechanistic explanation for improvements in event-based and spatial metrics such as CSI and FSS.

# 4. Ablation Experiment

## 4.1. Language Conditioning Strength Analysis

Figure 9 and Table 3 analyze the effect of motion-text conditioning under different classifier-free guidance (CFG) scales on two precipitation nowcasting datasets. CFG modu-

*Table 4.* Ablation study on temporal modeling and WCUB.

| Model | WCUB | Temporal Module | PSNR$\uparrow$ (dB) | SSIM$\uparrow$ | FVD$\downarrow$ |
|---|---|---|---|---|---|
| 2D VAE | $\checkmark$ | Temporal Shift Modules | **29.65** | **0.952** | **6.94** |
| 2D VAE | $\checkmark$ | Conv3D(OutBlock) | 29.10 | 0.945 | 8.03 |
| 2D VAE | $\checkmark$ | Conv3D ResBlock | 21.77 | 0.753 | 131.13 |
| 2D VAE | $\checkmark$ | $\times$ | 29.10 | 0.945 | 7.81 |
| 2D VAE | $\times$ | $\times$ | 28.78 | 0.943 | 9.18 |

lates the relative strength of high-level semantic constraints provided by textual motion descriptions during generation. Following classifier-free guidance, the guided velocity (or score) is computed as

$$\hat{v}_{\text{CFG}}(x_t, c) = (1 + s) \cdot v(x_t, c) - s \cdot v(x_t, \varnothing), \quad (11)$$

where $v(x_t, c)$ denotes the velocity prediction conditioned on motion text $c$, $v(x_t, \varnothing)$ is the unconditional prediction, and $s$ controls the strength of textual conditioning.

Across both datasets, incorporating motion descriptions consistently improves CSI and FSS metrics, indicating that textual guidance provides complementary semantic priors beyond visual observations. On the Swedish dataset, language conditioning yields a 40% gain in extreme precipitation skill (CSI at 6.3 mm/h: 0.0530 → 0.0746). On MRMS, improvements are more modest (CSI at 8 mm/h: +2.3%), as global metrics are less sensitive to high-level semantic constraints in this highly skewed distribution. Nevertheless, consistent FSS gains across both datasets (Swedish: 0.920 → 0.923; MRMS: 0.722 → 0.725) confirm that motion descriptions effectively regularize spatiotemporal structure and enhance extreme-event detection.

## 4.2. Temporal Modeling and WCUB Design

To isolate decoder-side temporal modeling, we fix the 2D VAE encoder without temporal compression and evaluate different decoder designs (Table 4). Under this setting, the proposed WCUB consistently improves spatial reconstruction over the baseline 2D VAE, and further gains are achieved by adding lightweight Temporal Shift Modules (TSM), yielding the best overall performance in PSNR, SSIM, and FVD. In contrast, directly introducing 3D convolutions or 3D residual blocks leads to unstable optimization and severe FVD degradation, indicating that aggressive spatiotemporal modeling is ineffective when the encoder lacks temporal compression.

## 4.3. Component Contribution Analysis

To disentangle the contributions of language conditioning and the WCUD decoder, we conduct a cross-ablation study by independently removing each component (Table 5). Language conditioning alone improves CSI$\geq$6.3 by 14.7% (0.0503→0.0577) and FSS from 0.905 to 0.917. WCUD alone reduces CRPS by 9.7% (0.1219→0.1101) and raises

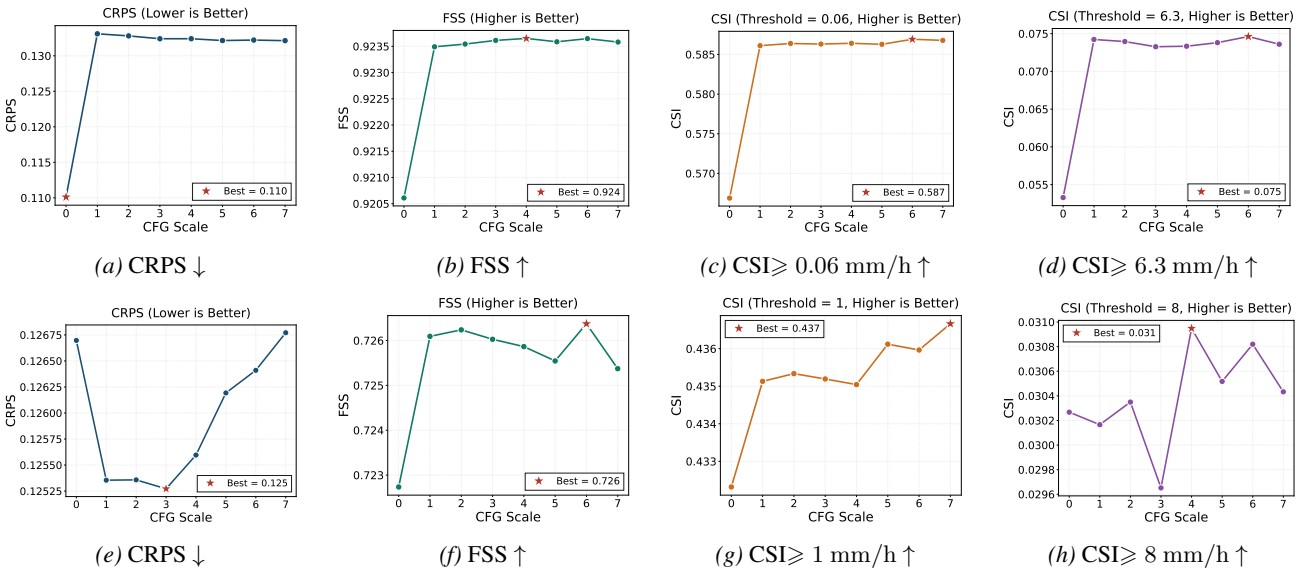

*Figure 9.* Effect of CFG scale on precipitation nowcasting performance. Top row: Swedish dataset. Bottom row: MRMS dataset. Event-based metrics (CSI) and spatial consistency (FSS) consistently improve with moderate CFG, while CRPS exhibits limited variation. When CFG is set to 0, the model degenerates to a purely visual, single-modality nowcasting baseline.

*Table 5.* Component ablation on the Swedish dataset. Language conditioning and WCUD contribute independently; their combination achieves the strongest overall performance.

| Language | WCUD | CRPS↓ | CSI≥0.06↑ | CSI≥6.3↑ | FSS↑ |
|----------|------|-------|-----------|----------|------|
| ✓ | ✓ | 0.1322 | **0.5869** | **0.0746** | **0.923** |
| ✗ | ✓ | **0.1101** | 0.5668 | 0.0530 | 0.920 |
| ✓ | ✗ | 0.1405 | 0.5736 | 0.0577 | 0.917 |
| ✗ | ✗ | 0.1219 | 0.5580 | 0.0503 | 0.905 |

FSS to 0.920. The full model achieves the best discriminative performance, with CSI≥6.3 reaching 0.0746 (+48.3% over the no-language, no-WCUD baseline) and FSS of 0.923. The slightly higher CRPS of the full model reflects a known trade-off: language conditioning concentrates the generative distribution toward specific motion patterns, improving spatial precision (CSI↑, FSS↑) at the cost of reduced ensemble diversity (CRPS↑). **Language conditioning is the primary driver of extreme-precipitation improvements, while WCUD provides complementary gains in reconstruction fidelity and spatial coherence.**

## 5. Conclusion

We presented LangPrecip, a language-aware multimodal framework that mitigates motion ambiguity in precipitation nowcasting by introducing meteorological motion semantics as explicit conditioning signals. Supported by LangPrecip-160K, the first large-scale radar-text paired dataset, and a semantically conditioned Rectified Flow formulation, Lang-Precip substantially improves temporal stability and extreme precipitation detection, achieving over 60% relative CSI

gains in heavy-rainfall detection compared to vision-only methods.

**Limitations.** Motion descriptions are generated using general-purpose vision-language models without meteorological specialization, which may overlook domain-specific dynamics. Additionally, generalization to diverse climate regimes (e.g., tropical convection, tropical cyclones, and monsoon systems) has not yet been systematically validated.

**Future Work.** Several directions remain open for exploration. First, incorporating structured meteorological signals such as NWP wind fields as complementary motion guidance alongside language descriptions could provide more physically explicit priors, particularly for systems where optical-flow-derived motion is noisy or ambiguous. Second, the promising interaction between pixel-level and semantic-level motion representations motivates a hierarchically motion-aware generative architecture, in which local displacement is modeled implicitly at lower network layers while language-derived semantic priors constrain system-level evolution at higher layers—allowing fine-grained dynamics and high-level semantic constraints to mutually reinforce across scales.

## Acknowledgements

This work was supported in part by the Science and Technology Program of Sichuan Province under Grant 2026NS-FSC0430, and in part by the Yibin Municipal Science and Technology Bureau under Grant 2025JC008.

## Impact Statement

This work aims to advance short-term precipitation nowcasting through language-guided machine learning methods. We identify the following potential societal impacts:

**Positive impacts:** Improved precipitation nowcasting has significant benefits for public safety and disaster management. Our method's enhanced performance on heavy-rainfall prediction (60% improvement in CSI at 80-minute lead time) could strengthen early warning systems for flash floods and severe weather events, potentially saving lives and reducing economic losses. The improved spatial detail preservation may benefit applications requiring high-resolution forecasts, such as urban flood management, aviation safety, and emergency response planning.

**Dataset contribution:** We will release LangPrecip-160K, a large-scale radar-text paired dataset curated from publicly available meteorological sources, to facilitate open research in multimodal weather forecasting.

**Limitations and responsible use:** Despite improvements, precipitation forecasting remains probabilistic and uncertain, particularly for extreme events. Our model's performance may vary across different geographic regions and climatic conditions beyond the Swedish and MRMS benchmarks tested. We emphasize that our system should complement, not replace, existing operational forecasting systems and human meteorological expertise. Users should interpret predictions with appropriate uncertainty awareness and validate performance in their specific deployment contexts before relying on forecasts for critical decision-making. The language-annotation approach introduces additional data requirements that may limit immediate deployability in data-scarce regions.

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

# A. Appendix

## A.1. Dataset details

### A.1.1. PROMPT TEMPLATE FOR RADAR ECHO VIDEO CAPTIONING

We use the following prompt template to guide the language model in generating professional meteorological descriptions for radar echo videos:

---

**Prompt:** As a professional radar meteorology analyst, generate a detailed expert annotation for this radar echo video. This analysis will be used for video captioning and description generation. Accurately describe only what is observed in the video.

**Analysis Framework:**

1. **Spatial Distribution:** Describe where echoes appear in the image and the main concentration regions.

2. **Morphological Characteristics:** Analyze geometric shapes and precipitation-area morphology, including:

   - Shape types: linear, banded, cellular (blob-like), arc-shaped, spiral, etc.
   - Precipitation structures: core region, gradient zone, echo band, echo wall, bow echo, hook echo, etc.
   - Boundary features: edge sharpness, boundary regularity, gradient characteristics
   - Internal structure: echo intensity distribution, embedded cores, voids or weak-echo regions

3. **Temporal Evolution:**

   - Motion characteristics: Describe direction and speed type (translation/rotation/deformation)
   - Area changes: Describe trend indicators and patterns
   - Morphological evolution: Describe changes over time in shape, intensity, and boundaries

4. **Stability and Consistency:** Evaluate regularity based on observed patterns.

**Output Requirements:**

- Use professional terminology while keeping the text clear and readable for a video caption

- Provide concrete descriptions based on visual observations

- Make the analysis comprehensive and logically structured to form a coherent description

- Balance morphology and dynamic analysis; keep the total length within 200–300 words

- Use fluent, natural English and avoid overly technical wording that reduces readability

- End with a brief phenomenon summary (3–4 sentences) that summarizes the main features and trends; the text should not include any heading

---

A.1.2. ANNOTATION INTERFACE AND QUALITY CONTROL PIPELINE

As shown in Figure 10, the annotation pipeline proceeds in three stages. First, Qwen-VL generates a full narrative motion description from the radar echo sequence using the prompt template above. Second, a lightweight LLM extracts and highlights key meteorological attributes—propagation direction, intensity evolution, structural changes such as expansion or contraction, and rotational dynamics—directly within the web interface. Third, trained meteorological experts review each sample item by item, verifying physical plausibility and flagging inconsistencies for correction. This pipeline is applied exclusively to the training split; the test set remains entirely unannotated, ensuring that evaluation reflects real-world inference conditions where no curated descriptions are available.

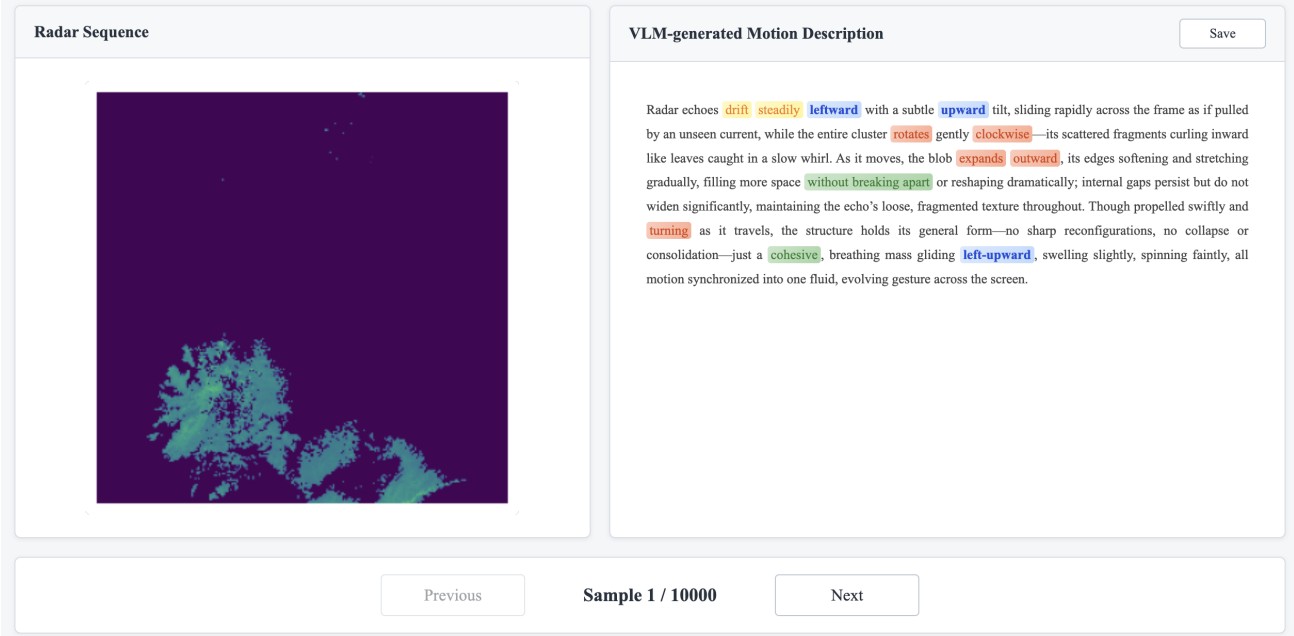

*Figure 10.* Annotation interface for human expert review. The left panel displays the radar reflectivity sequence; the right panel shows the VLM-generated motion description with key meteorological terms (propagation direction, rotation, structural changes) highlighted for rapid expert verification. Experts inspect each sample sequentially and flag descriptions requiring correction before saving.

A.1.3. SWEDISH DATASET OVERVIEW.

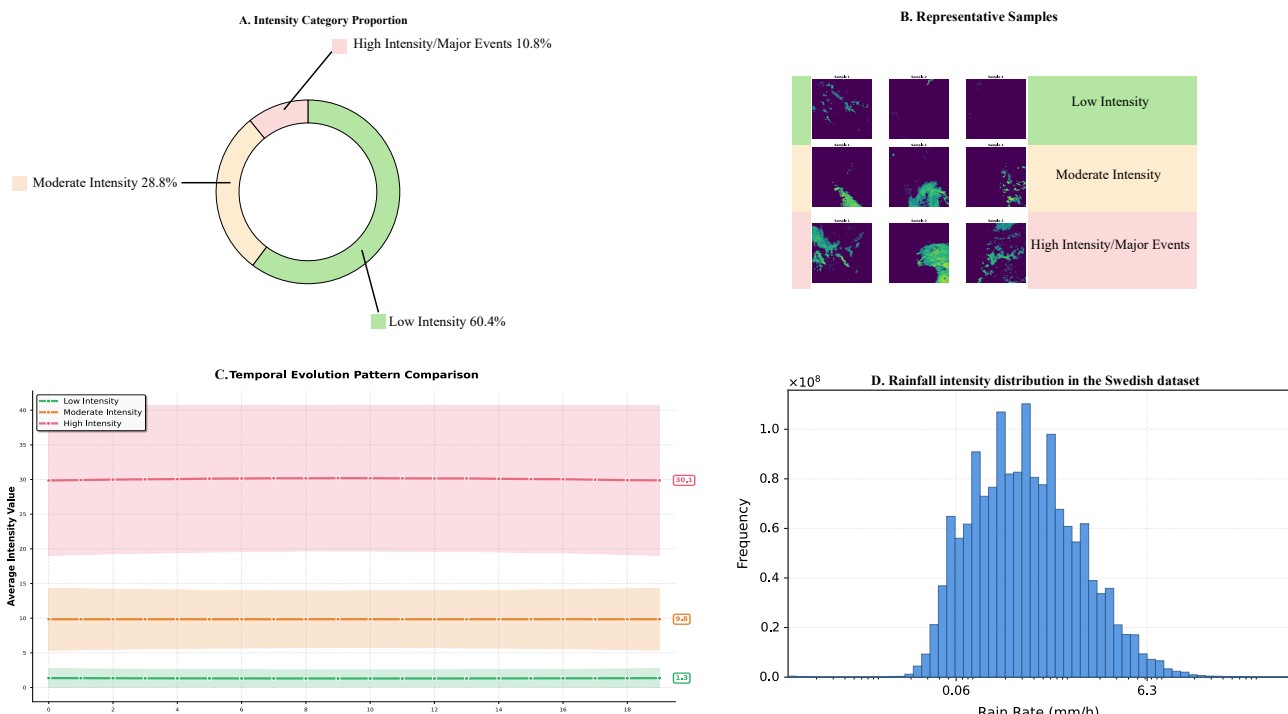

*Figure 11.* Statistical characteristics of the Swedish precipitation dataset. (A) Proportion of precipitation intensity categories, including low-, moderate-, and high-intensity events. (B) Representative radar samples for each intensity category, illustrating increasing spatial coverage and structural complexity with higher intensity. (C) Temporal evolution of average precipitation intensity across categories over the 100-minute sequences, showing distinct magnitude levels and stable intensity trends for low-, moderate-, and high-intensity events. (D) Rainfall intensity distribution of the Swedish dataset (randomly sampled 200k frames).

We construct a large-scale Swedish radar precipitation dataset covering a continuous period from 2017 to 2021. Each event consists of a fixed-length radar echo sequence sampled at 5-minute intervals, forming short-term spatiotemporal observations for precipitation nowcasting. All sequences are spatially cropped and temporally aligned following standard preprocessing protocols to ensure consistency across samples. To characterize the inherent heterogeneity of precipitation intensity, we stratify the dataset into three categories based on average rainfall intensity: low intensity, moderate intensity, and high intensity/major precipitation events. This stratification reveals a pronounced imbalanced distribution, as shown in Figure 11(A), where low-intensity events dominate the dataset, while extreme precipitation events constitute only a small fraction of samples.

Representative radar snapshots in Figure 11(B) illustrate clear structural differences across intensity regimes. Low-intensity events are characterized by sparse and fragmented echoes, moderate-intensity events exhibit more organized precipitation structures, and high-intensity events contain compact, high-reflectivity cores with complex spatial morphology, often associated with severe weather systems. Despite large differences in magnitude, the temporal evolution patterns within each category remain relatively stable over short lead times, as shown in Figure 11(C). The average intensity for each regime exhibits limited fluctuation over time, suggesting that short-term precipitation evolution is primarily governed by coherent advection and gradual intensity changes. This property supports nowcasting over short lead times while simultaneously highlighting the intrinsic difficulty of accurately predicting rare extreme events due to their scarcity and structural complexity.

The rainfall intensity distribution in Figure 11(D) shows that the Swedish dataset spans from light to heavy precipitation, with a relatively concentrated distribution compared to typical highly skewed rainfall datasets. A substantial proportion of observations lies above the light-rain threshold of 0.06 mm/h (left dashed line), while a noticeable fraction extends beyond the heavy-rainfall threshold of 6.3 mm/h (right dashed line). Although low-intensity events comprise the majority of samples, moderate and heavy precipitation events are sufficiently represented to support comprehensive model evaluation across intensity regimes, especially for threshold-based heavy-precipitation forecasting.

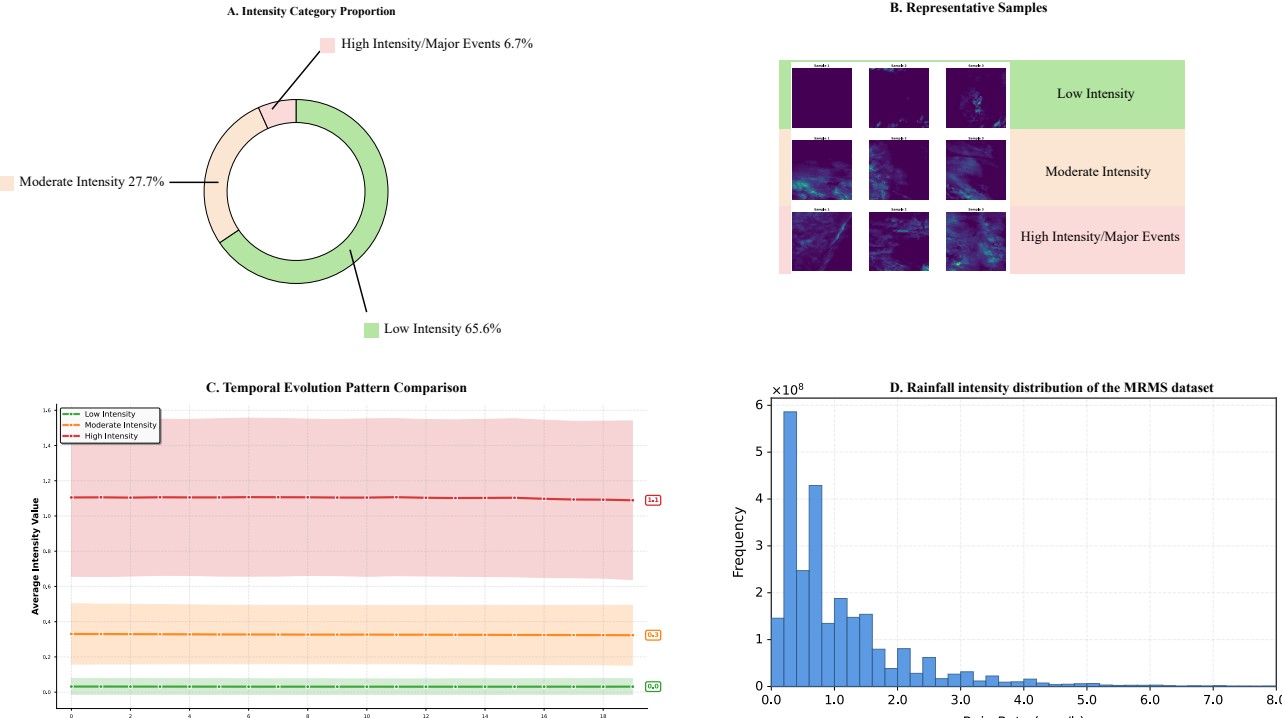

*Figure 12.* Statistical characteristics of the MRMS precipitation dataset. (A) Distribution of precipitation intensity categories, including low-, moderate-, and high-intensity (major) events. (B) Representative radar samples for each category, illustrating the increasing spatial coverage and structural complexity associated with higher precipitation intensity. (C) Temporal evolution of average precipitation intensity across categories over the 100-minute sequences, showing clear separation in magnitude and distinct intensity levels for extreme events. (D) Rainfall intensity distribution of the MRMS dataset (randomly sampled 200k frames).

### A.1.4. MRMS DATASET OVERVIEW.

In addition to the Swedish dataset, we also construct an MRMS-based precipitation nowcasting dataset derived from the Multi-Radar/Multi-Sensor (MRMS) system, which provides high-resolution, continental-scale radar observations over the United States. This dataset exhibits substantially different climatological and structural characteristics compared to the Swedish radar data.

Representative radar snapshots in Figure 12(B) reveal clear intensity-dependent differences in radar reflectivity distributions. Low-intensity events are dominated by uniformly weak returns, whereas moderate- and high-intensity samples exhibit higher reflectivity levels with increased spatial variability. In high-intensity MRMS events, reflectivity values are more unevenly distributed across the spatial domain, indicating stronger local contrasts in precipitation intensity. The temporal evolution statistics in Figure 12(C) further highlight systematic differences between MRMS and Swedish precipitation. While the average intensity within each regime remains relatively stable over short lead times, MRMS exhibits higher absolute intensity levels and wider variability, particularly for high-intensity events. This suggests stronger nonlinearity and greater intrinsic uncertainty in short-term precipitation evolution, posing additional challenges for accurate nowcasting.

The rainfall intensity distribution in Figure 12(D), estimated from 200k randomly sampled radar frames, exhibits a strongly right-skewed pattern in the MRMS dataset. The majority of observations concentrate at low rainfall intensities below 1.0 mm/h (marked by the red dashed line), with frequency exceeding $6 \times 10^8$ for the lowest intensity bin. The distribution shows a rapid exponential decay toward higher intensities, with progressively fewer occurrences beyond 1.0 mm/h. This distribution highlights the relative rarity of intense precipitation events, with the dominance of low-intensity observations and the scarcity of extreme rainfall introducing inherent imbalance challenges for nowcasting model training and evaluation.

**A.2. Comparative Analysis: Radar Echo Sequences vs. Natural Videos**

In this section, we provide a detailed analysis of the fundamental differences between the LangPrecip-160K precipitation radar echo dataset and conventional video datasets used in general computer vision, such as UCF101[1], Kinetics[2], Panda-70M[3], or large-scale datasets like OpenSora[4] and Wan Video[5] employed for training modern video generation models. Understanding these physical and statistical disparities is crucial for explaining why standard video generation architectures, such as visual VAEs, perform poorly on radar echo forecasting tasks.

A.2.1. PHYSICAL FIDELITY VS. VISUAL REALISM

The primary objective of conventional video generation is typically perceptual realism, i.e., producing visually coherent outputs that align with human sensory expectations. Within this paradigm, minor pixel misalignments or semantic hallucinations are generally tolerable as long as the overall appearance remains natural and plausible.

In contrast, radar echo forecasting is fundamentally a task of physical data fidelity. Each pixel in a radar echo map corresponds to a quantitative physical measurement of rainfall intensity (rain rate), where even small spatial displacements or intensity errors can directly compromise the reliability of downstream warning and decision-making systems.

This fundamental mismatch motivates our proposal of the Wavelet Consistency Unfolding Decoder (WCUD). By explicitly modeling the underlying physical degradation processes, WCUD mitigates the smoothing and blurring artifacts that are prevalent in standard generative architectures, thereby preserving fine-grained intensity structures and spatial consistency critical for physically meaningful radar echo prediction.

A.2.2. STRUCTURAL SPARSITY AND EXTREME DISTRIBUTIONAL SKEW

Conventional video data typically exhibit dense spatial information with rich semantic textures distributed across most pixels.

In contrast, precipitation radar sequences are characterized by pronounced spatial sparsity, where the majority of pixels correspond to zero values representing no-rain regions. At the same time, localized precipitating areas exhibit sharply defined boundaries and intricate internal core structures. Accurately modeling this coexistence of vast empty regions and highly concentrated, high-intensity patterns poses a significant challenge for standard generative frameworks.

Moreover, precipitation data follow an extremely long-tailed distribution. In our Swedish and MRMS datasets, heavy rainfall events with substantial societal impact account for only 10.8% and 6.7% of all samples, respectively. General-purpose generative models trained on natural video datasets tend to regress toward the climatological mean, which severely limits their ability to capture the dynamic evolution of such rare but critical extreme events.

A.2.3. DOMAIN-SPECIFIC MOTION SEMANTICS AND CONSTRAINED DYNAMICS

In conventional video datasets, natural language descriptions typically refer to high-level, macroscopic behaviors, such as "a dog running on the grass," without imposing strict constraints on the underlying motion dynamics.

In contrast, within LangPrecip-160K, textual descriptions are defined as explicit kinematic priors. The generated annotations—such as advection direction, cyclonic rotation, and convergence/divergence patterns—directly correspond to well-defined meteorological dynamical processes.

As a result, text in our framework functions not merely as a content-level instruction, but as a physically grounded constraint signal. When the historical observation window is limited (typically to only four frames), these motion-aware textual priors effectively reduce future trajectory ambiguity, guiding the model toward dynamically consistent and physically plausible

---

[1]Soomro K, Zamir A R, Shah M. Ucf101: A dataset of 101 human actions classes from videos in the wild[J]. arXiv preprint arXiv:1212.0402, 2012.

[2]Kay W, Carreira J, Simonyan K, et al. The kinetics human action video dataset[J]. arXiv preprint arXiv:1705.06950, 2017.

[3]Chen T S, Siarohin A, Menapace W, et al. Panda-70m: Captioning 70m videos with multiple cross-modality teachers[C]//Proceedings of the IEEE/CVF Conference on Computer Vision and Pattern Recognition. 2024: 13320-13331.

[4]Zheng Z, Peng X, Yang T, et al. Open-sora: Democratizing efficient video production for all[J]. arXiv preprint arXiv:2412.20404, 2024.

[5]Wan T, Wang A, Ai B, et al. Wan: Open and advanced large-scale video generative models[J]. arXiv preprint arXiv:2503.20314, 2025.

evolutions.

## A.3. Denoise Model

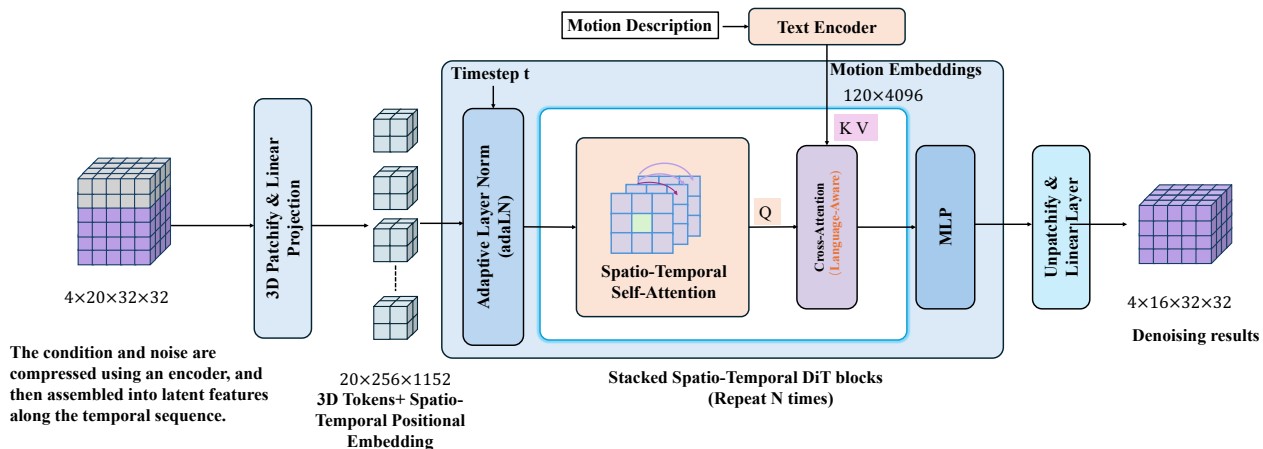

*Figure 13.* Denoising network architecture

The VAE encoder compresses high-dimensional radar observations into a compact latent space while preserving spatial structure. However, VAE latents optimized solely for reconstruction lack explicit physical constraints, potentially leading to ambiguous motion representations that accumulate errors in multi-step forecasting. To address this limitation, we introduce natural language as explicit semantic guidance to constrain the learned dynamics. By injecting textual motion descriptions into the denoising process, the model is guided to follow high-level physical evolution patterns (e.g., propagation direction and intensity trends) rather than relying on ambiguous visual features alone. This language-guided approach enables both faithful reconstruction and physically consistent forecast trajectories.

Our denoising network is a Transformer-based spatiotemporal model that parameterizes the Rectified Flow velocity field. As illustrated in Figure 13, we adopt a Diffusion Transformer (DiT) backbone operating on 3D latent representations, where the temporal dimension is explicitly modeled alongside spatial dimensions. The input latent variables are first tokenized via a 3D patchify operation followed by linear projection, then augmented with learnable spatiotemporal positional embeddings to preserve the inherent spatiotemporal structure. The core network consists of multiple stacked spatiotemporal DiT blocks.

Within each DiT block, features are first modulated by adaptive layer normalization (adaLN) conditioned on the diffusion timestep $t$. Spatiotemporal self-attention then captures long-range dependencies across both spatial and temporal dimensions, enabling the model to maintain spatial consistency within frames and temporal coherence across future evolution.

Following self-attention, a cross-attention mechanism injects semantic motion constraints into the denoising dynamics. In this mechanism, latent tokens serve as queries, while text-derived motion embeddings provide keys and values. This design allows semantic information to directly modulate the velocity field, constraining latent trajectories toward physically plausible precipitation evolution. The processed tokens are then passed through a feed-forward network, and finally an unpatchify operation reshapes them back to the original latent dimensionality to produce the denoised latent output.

**Inference Pipeline and Causal Consistency.** At inference time, the complete pipeline proceeds as follows: (1) the four most recent observed radar frames ($t \leq t_0$) are fed to the VLM, which generates a natural-language motion description; (2) the description is encoded via T5-XXL and injected into the denoising network as the language condition; (3) the model generates future precipitation frames conditioned on both the historical radar context and the motion description. No future frame information is involved at any stage—causal consistency is strictly guaranteed by construction. The entire pipeline, including VLM captioning, averages 5.4 s per sample on a single RTX 4090, well within the 5-minute operational radar update cycle.

**A.4. Precipitation nowcasting performance: Language-guided motion-aware vs. vision-only approaches**

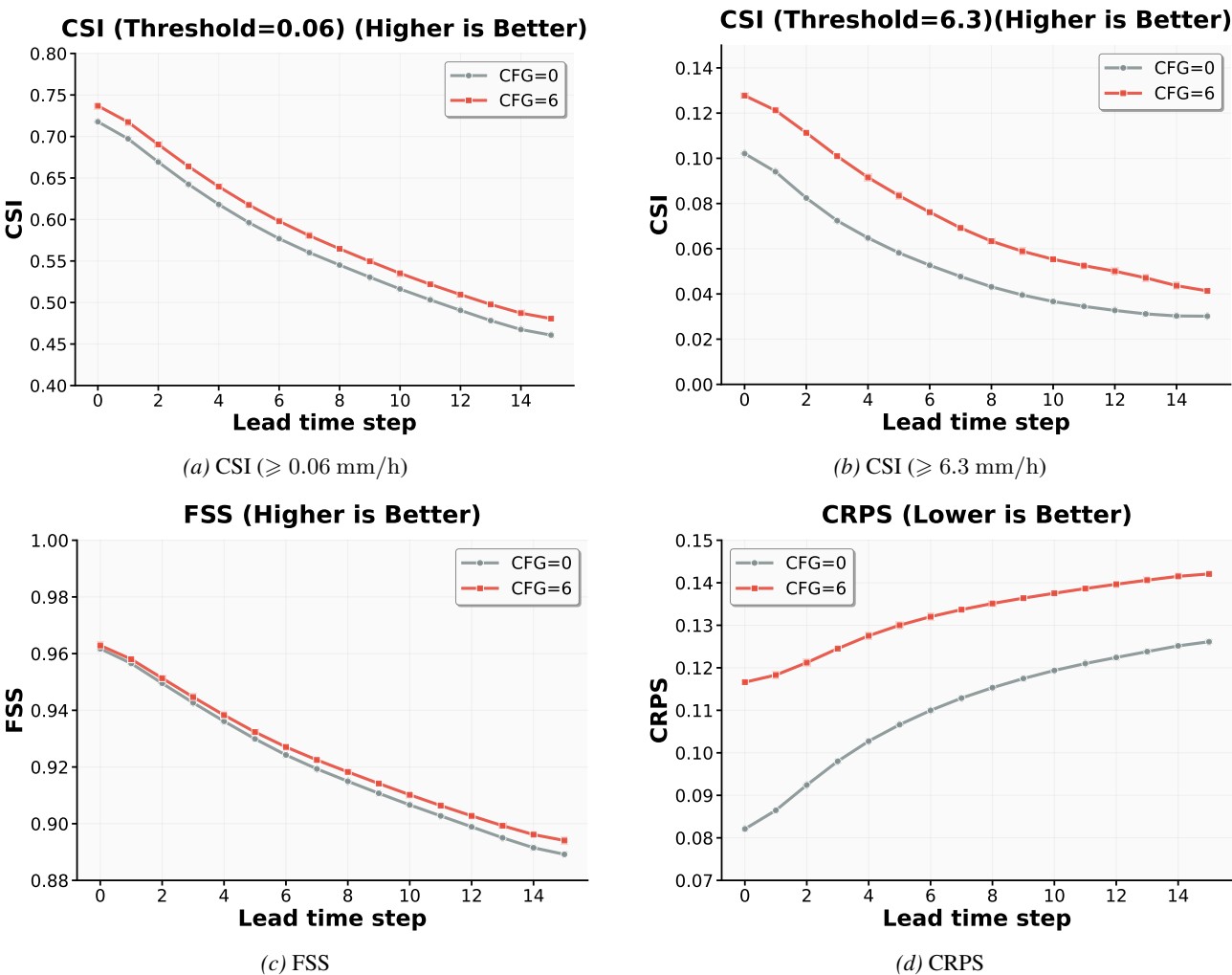

*Figure 14.* Temporal performance evolution on the Swedish dataset across lead times (5–80 minutes) for models with and without textual motion guidance (CFG=0 denotes vision-only). Results are reported in terms of CSI, FSS, and CRPS.

A.4.1. LANGUAGE-GUIDED VS. VISION-ONLY: SWEDISH DATASET

On the Swedish dataset, incorporating textual motion guidance consistently improves CSI across both light- and heavy-precipitation thresholds throughout the forecasting horizon. The improvement becomes more pronounced at longer lead times, particularly for heavy precipitation, indicating that semantic motion constraints help preserve event detectability as temporal uncertainty accumulates. Similar gains are observed for FSS, further suggesting enhanced spatial consistency and more coherent precipitation structures over time. Notably, the advantages of textual guidance are evident from the initial prediction stage and continue to increase with lead time, validating the consistent contribution of motion semantic constraints to long-term nowcasting stability.

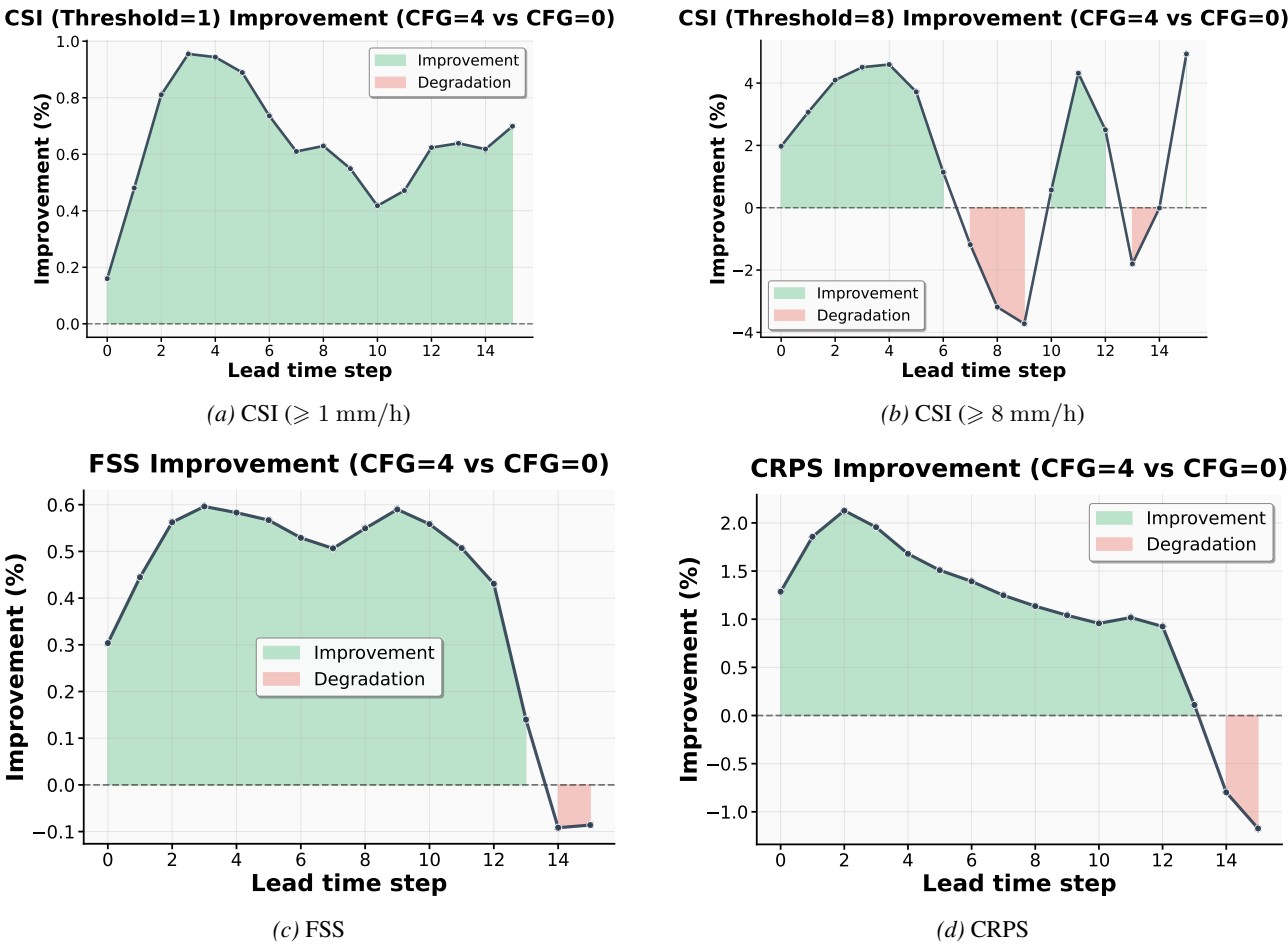

*Figure 15.* Relative performance improvement of textual motion guidance (CFG=4 vs CFG=0) on the MRMS dataset over 16 lead time steps (5–80 minutes, 5-minute intervals). Positive values indicate performance gains over the vision-only baseline for (a) CSI ($\geq 1$ mm/h), (b) CSI ($\geq 8$ mm/h), (c) FSS, and (d) CRPS metrics.

### A.4.2. LANGUAGE-GUIDED VS. VISION-ONLY: MRMS DATASET

The relatively modest performance gains observed on the MRMS dataset can be partly attributed to its statistical characteristics. As shown in Figure 12(D), MRMS exhibits a highly right-skewed, heavy-tailed rainfall intensity distribution, where weak precipitation dominates the majority of samples and intense rainfall events are comparatively rare. Under these conditions, short-term forecasts are strongly constrained by local observations and immediate dynamics, limiting the potential for additional semantic priors to improve performance at early lead times. Nevertheless, textual motion guidance still yields consistent improvements at longer lead times and under heavy-precipitation thresholds. As visual constraints gradually weaken with increasing forecast horizons, the semantic constraints imposed by motion text—encoding coarse propagation patterns and overall evolution trends—regularize the latent dynamics, mitigating cumulative trajectory drift and stabilizing spatiotemporal structures.

Despite the challenging distributional characteristics of MRMS, our approach demonstrates that textual motion guidance provides meaningful performance gains where they matter most. First, at longer forecast horizons beyond 40 minutes (lead time step 8+), where visual information becomes less reliable, CRPS improvements reach 2% at 20 minutes and continue to grow with lead time. Second, for extreme precipitation events assessed by CSI at the 8 mm/h threshold, which are critical for early warning systems, improvements reach up to 5% at mid-range lead times (25–35 minutes). The consistent positive trends across CSI, FSS, and CRPS metrics validate that semantic motion priors complement rather than replace visual information, offering a robust regularization mechanism that becomes increasingly valuable as prediction uncertainty grows. These findings suggest that textual motion guidance is particularly beneficial for operational nowcasting systems that prioritize long-lead forecasts and extreme event detection.

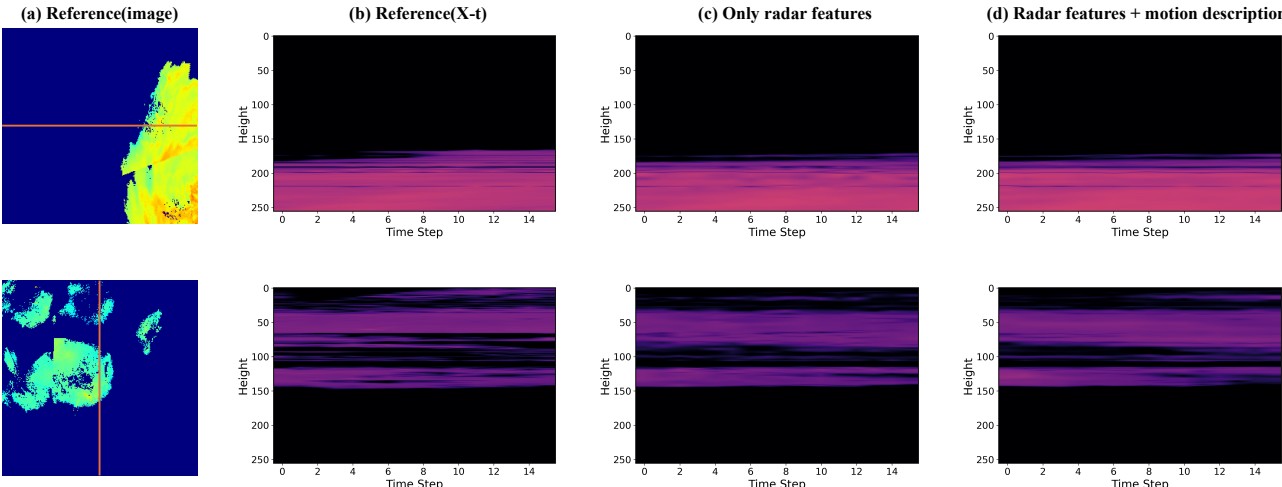

*Figure 16.* Space-time slices of precipitation sequences along a spatial profile (orange line in (a)). (a) Radar frame showing the selected profile location. (b) Ground truth precipitation evolution. (c) Predictions without text guidance (vision-only). (d) Predictions with text guidance. All panels show precipitation evolution along the profile over 16 time steps.

### A.4.3. LANGUAGE-GUIDED VS. VISION-ONLY: SPACE-TIME SLICE ANALYSIS

To examine the temporal consistency and physical plausibility of the generated nowcasting results, we employ space-time $(X - t)$ slice visualization, as shown in Figure 16. By tracking radar reflectivity values along a fixed spatial transect over time, this representation projects high-dimensional spatiotemporal forecasts into a two-dimensional space, enabling fine-grained inspection of temporal evolution patterns.

As illustrated in Figure 16, predictions conditioned solely on radar observations exhibit pronounced temporal inconsistencies. These artifacts manifest as abrupt intensity changes between adjacent time steps, horizontal banding patterns, and unnaturally sharpened precipitation boundaries, indicating that vision-only models struggle to maintain coherent long-range temporal dynamics and tend to generate frames that are locally consistent but globally unstable. In contrast, incorporating textual motion descriptions as conditional guidance substantially improves temporal smoothness and structural coherence. The language-guided $X - t$ slices exhibit more continuous intensity transitions and smoother spatial gradients that align more closely with ground-truth observations, while effectively suppressing spurious artifacts.

These observations demonstrate that semantic motion cues provided by the text modality impose meaningful global constraints on the generation process, enhancing spatiotemporal consistency. The results validate that cross-modal conditioning effectively compensates for the limitations of purely visual representations in modeling long-term temporal dependencies, leading to more physically plausible precipitation evolution trajectories.

### A.4.4. MOTION REPRESENTATION COMPARISON

*Table 6.* Comparison of different motion conditioning strategies on the Swedish dataset (with WCUD). Language conditioning outperforms pixel-level alternatives on all discriminative metrics.

| Motion Representation | CRPS↓ | CSI≥0.06↑ | CSI≥6.3↑ | FSS↑ |
| --- | --- | --- | --- | --- |
| Language (ours) | 0.1322 | **0.5869** | **0.0746** | **0.923** |
| Implicit embedding | 0.1101 | 0.5668 | 0.0530 | 0.920 |
| Optical flow | **0.1093** | 0.5708 | 0.0529 | 0.921 |

To verify that the gains from language conditioning are not simply attributable to any form of motion information, we compare it against two pixel-level alternatives: optical flow extracted from consecutive radar frames, and a motion embedding learned implicitly from radar sequences. As shown in Table 6, optical flow and implicit embedding achieve nearly identical CSI≥6.3 (0.0529 vs. 0.0530), indicating that pixel-level motion representations provide limited conditional guidance. In contrast, language conditioning raises CSI≥6.3 to 0.0746 (+41.0% over optical flow). The slightly lower CRPS of pixel-level

methods reflects their more concentrated predictive distributions rather than superior forecasting skill. These results confirm that language compresses radar dynamics into a structured high-level motion prior that is qualitatively more effective than direct pixel-level motion representations for guiding precipitation generation.

### A.4.5. CFG ROBUSTNESS UNDER DIFFERENT DESCRIPTION QUALITIES

To analyze sensitivity to description quality, we design three groups of experiments varying the language input while keeping all other settings fixed (CFG scale 2–6, with WCUD): **Full** uses the complete LLM-generated descriptions as in the main paper; **Simplified** reduces each description to a single short phrase capturing core motion semantics (e.g., "radar echoes moving toward the upper right and gradually weakening"); **Reversed** replaces the motion direction in the full description with its exact opposite.

*Table 7.* CFG sensitivity under Full, Simplified, and Reversed descriptions on the Swedish dataset (with WCUD).

| CFG | CRPS$\downarrow$ | FSS$\uparrow$ | CSI$\geq$0.06$\uparrow$ | CSI$\geq$6.3$\uparrow$ |
|---|---|---|---|---|
| **Full** | | | | |
| 2 | 0.1328 | 0.9235 | 0.5864 | 0.0740 |
| 4 | 0.1324 | 0.9237 | 0.5864 | 0.0733 |
| 6 | 0.1322 | 0.9237 | 0.5869 | 0.0746 |
| **Simplified** | | | | |
| 2 | 0.1339 | 0.9235 | 0.5865 | 0.0738 |
| 4 | 0.1350 | 0.9234 | 0.5870 | 0.0744 |
| 6 | 0.1367 | 0.9230 | 0.5875 | 0.0737 |
| **Reversed** | | | | |
| 2 | 0.8569 | 0.8642 | 0.5890 | 0.0256 |
| 4 | 1.3433 | 0.7816 | 0.5667 | 0.0186 |
| 6 | 1.3498 | 0.7622 | 0.5507 | 0.0179 |

As shown in Table 7, Full descriptions improve monotonically with CFG strength. Simplified descriptions achieve CSI$\geq$6.3 of 0.0737, near-identical to Full (0.0746), demonstrating that coarse directional semantics suffice to maintain near-optimal performance. Reversed descriptions collapse even at CFG=2 (CRPS: 0.8569, FSS: 0.8642), confirming the model genuinely responds to language semantics rather than treating descriptions as noise. In practice, VLM-generated descriptions are derived from the same observed radar frames, making directional reversal virtually impossible under normal operating conditions.

## A.5. Effect of VAE Architecture

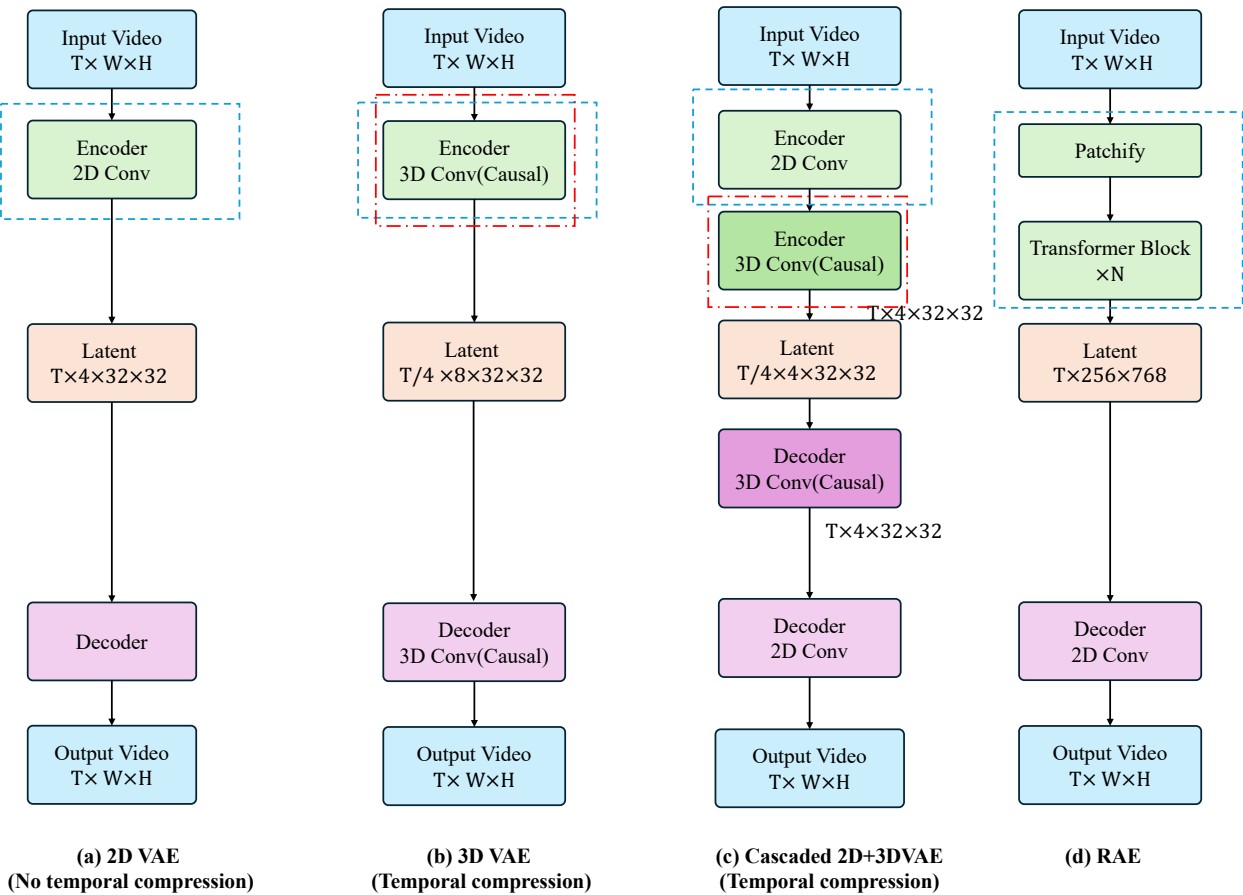

*Figure 17.* Architectural comparison of spatiotemporal autoencoders with different compression strategies. Blue dashed boxes indicate spatial compression stages, while red dashed boxes indicate temporal compression stages. Different strategies lead to distinct forms of information encoding in the latent representation.

From an architectural perspective, the key difference among these autoencoder variants lies in whether temporal compression is performed and how spatial compression is implemented (Figure 17).

2D VAEs compress only the spatial dimensions (blue dashed boxes), maintaining frame-by-frame encoding and decoding. Their latent variables retain the full temporal length ($16 \times 4 \times 32 \times 32$), making the reconstruction objective essentially frame-by-frame pixel recovery.

In contrast, 3D VAEs perform temporal downsampling during encoding through causal 3D convolutions (red dashed boxes), compressing the temporal dimension from 16 to 4 frames (yielding $4 \times 8 \times 32 \times 32$ latents). The decoder must then recover longer output sequences from these shorter latent representations, effectively requiring the reconstruction process to incorporate temporal prediction and interpolation components.

Cascaded 2D+3D schemes[6] further impose dual bottlenecks on both spatial and temporal dimensions (producing latents such as $4 \times 4 \times 32 \times 32$), resulting in a narrower information pathway where errors can more easily accumulate and amplify during decoding.

RAE[7] employs global semantic encoding via patchify and Transformer operations, producing token sequence representations (e.g., $T \times 256 \times 768$ where $T$ is the number of temporal tokens). Its learning objective prioritizes semantic consistency over pixel-wise reversible reconstruction (Figure 17).

---

[6]Zheng Z, Peng X, Yang T, et al. Open-sora: Democratizing efficient video production for all. arXiv preprint arXiv:2412.20404, 2024.

[7]Zheng B, Ma N, Tong S, et al. Diffusion transformers with representation autoencoders. arXiv preprint arXiv:2510.11690, 2025.

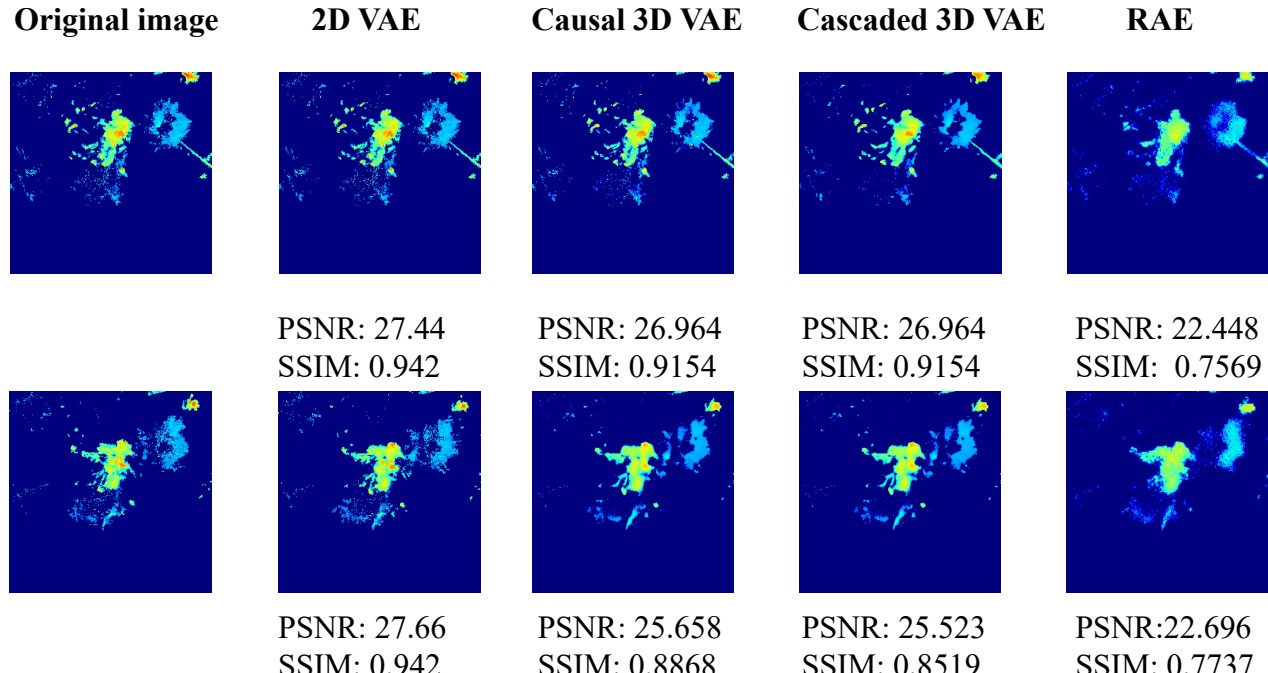

*Figure 18.* Comparison of visual effects reconstructed by different VAE architectures

*Table 8.* Comparison of different VAE architectures for radar sequence reconstruction. All models are trained with L1 reconstruction loss only.

| Model | Latent Shape ($T \times C \times H \times W$) | PSNR (dB) | SSIM |
|---|---|---|---|
| 2D VAE (No temporal compression) | $16 \times 4 \times 32 \times 32$ | **27.27** | **0.9328** |
| Fully Causal 3D VAE | $4 \times 8 \times 32 \times 32$ | 26.02 | 0.9139 |
| Cascaded 3D VAE | $4 \times 4 \times 32 \times 32$ | 24.37 | 0.8415 |
| RAE | $256 \times 768$ | 21.54 | 0.7516 |

### A.5.1. IMPACT OF AUTOENCODER DESIGN ON RECONSTRUCTION QUALITY

The quantitative results clearly reflect the trade-off between temporal compression and reconstruction quality imposed by the information bottleneck (Table 8). The 2D VAE without temporal compression achieves the highest PSNR/SSIM (27.27 dB / 0.9328), indicating that preserving complete temporal resolution is crucial for pixel-level fidelity in tasks sensitive to local texture and intensity gradients, such as precipitation radar echo reconstruction. The Fully Causal 3D VAE with temporal compression shows reduced performance (26.02 dB / 0.9139), as temporal downsampling forces the model to compress multi-frame details into shorter latent representations, sacrificing spatial fidelity. The Cascaded 3D VAE with dual spatial-temporal compression further degrades to 24.37 dB / 0.8415, confirming that cascaded compression incurs more severe information loss. RAE exhibits the lowest PSNR/SSIM (21.54 dB / 0.7516), consistent with its training objective that prioritizes global semantic representation over pixel-wise reconstruction accuracy, making it less suitable for precipitation nowcasting applications that demand high spatial precision.

The visualization results corroborate these quantitative findings (Figure 18). 2D VAE reconstructions better preserve the shape and fine structure at the core and boundaries of strong echoes, with background noise and weak echo regions that more closely match the original images. In contrast, causal 3D VAEs and cascaded 3D VAEs exhibit more pronounced smoothing and diffusion at echo boundaries, with local peaks and small-scale features being suppressed—reflecting the averaging effect that occurs when temporal compression represents multiple frames through fewer temporal channels. RAE reconstructions show stronger semantic approximation: while the approximate locations of main echo clusters are preserved, intensity distributions and fine-grained textures are significantly distorted, resulting in substantially lower PSNR/SSIM values.

From a mechanistic perspective, these performance differences stem from several key factors. First, temporal compression fundamentally maps high-frequency temporal variations across multiple frames onto a much shorter latent temporal axis, substantially increasing the information load that each latent time step must encode. This creates a pronounced representational bottleneck, forcing the model to preferentially preserve low-frequency and more compressible spatiotemporal structures—such as large-scale echo patterns and slowly evolving trends—while inevitably discarding high-frequency details associated with rapid dynamics, including sharp echo boundaries, localized intensification, and dissipation events.

Second, although fully causal 3D convolutions structurally enforce strict temporal causality and consistency, they simultaneously restrict the use of bidirectional temporal context during encoding and decoding. Since representations at any given time step cannot access future frames, the model cannot leverage symmetric temporal context to correct local spatial reconstruction errors. In reconstruction tasks requiring integration of information from both past and future frames to recover fine-grained spatial structures, this unidirectional constraint further degrades spatial fidelity, manifesting as blurred boundaries and smoothed details.

Third, cascaded temporal compression compounds two sequential irreversible projections: spatial compression followed by temporal compression. Spatial downsampling already induces loss of local texture and intensity information; subsequent temporal compression further collapses the dynamic variation dimension, causing reconstruction errors to accumulate and amplify along the decoding pathway. This compounded bottleneck not only reduces the effective information density within latent representations but also intensifies interpolation and averaging effects during decoding, leading to increased suppression of weak echoes and more pronounced blurring of strong-echo boundaries.

In contrast, RAE employs a contrastive representation learning objective (DINO) that prioritizes global semantic alignment and cross-view invariance over pixel-wise invertible reconstruction. This objective inherently relaxes constraints on local intensity distributions and fine-scale spatial structures. Consequently, while the model may preserve large-scale echo morphology and overall spatial layout, it struggles to accurately recover local extrema, sharp gradients, and fine textures in precipitation intensity fields. For precipitation nowcasting applications demanding high spatial precision, these reconstruction characteristics render RAE less favorable compared to approaches that maintain stronger pixel-level fidelity.

### A.6. Wavelet Consistency Unfolding Block Pseudocode

---

**Algorithm 1** Iterative Reconstruction with Data Consistency and Wavelet Prior

---

**Input:** low-resolution observation $y$, iterations $T$, step sizes $\{\eta_t\}$, thresholds $\{\tau_t\}$
**Output:** high-resolution reconstruction $\hat{x}$
Initialize $x^{(0)} \leftarrow U_s(y)$     *(upsampling initialization)*
**for** $t = 0$ to $T - 1$ **do**
   *Data consistency step*
   $\hat{y}^{(t)} \leftarrow D_s\big(x^{(t)} * k\big)$
   $r \leftarrow y - \hat{y}^{(t)}$
   $z^{(t+1)} \leftarrow x^{(t)} + 2\eta_t \, D_s^\top(r) * k^\top$
   *(often $k^\top$ is absorbed into $D_s^\top$, i.e., $z^{(t+1)} \approx x^{(t)} + 2\eta_t D_s^\top(r)$)*
   *Wavelet prior step*
   $(L, H_h, H_v, H_d) \leftarrow W\big(z^{(t+1)}\big)$
   $H_h \leftarrow \mathrm{shrink}(H_h, \tau_t)$
   $H_v \leftarrow \mathrm{shrink}(H_v, \tau_t)$
   $H_d \leftarrow \mathrm{shrink}(H_d, \tau_t)$
   $x^{(t+1)} \leftarrow W^\top(L, H_h, H_v, H_d)$
**end for**
**return** $\hat{x} \leftarrow x^{(T)}$

---

## A.7. Training Hyperparameter

*Table 9.* Hyperparameter settings for training the proposed LangPrecip model and the 2D VAE.

| (A) LangPrecip (Rectified Flow in latent space) | | |
|---|---|---|
| **Item** | **Symbol / Option** | **Value** |
| Input frames | $T_{\text{in}}$ | 20 (4 condition + 16 noise) |
| Output frames | $T_{\text{out}}$ | 16 |
| Original resolution | $H \times W$ | $256 \times 256$ |
| Latent downsampling factor | — | $\times 8$ |
| Latent resolution | — | $32 \times 32$ |
| Latent channels | $C_z$ | *4* |
| Conditioning | — | Radar history + motion text |
| Text encoder | — | *T5 XXL*(Chung et al., 2024) |
| Text token length | $L$ | *120* |
| Backbone | — | *DTCA*(Li et al., 2024) |
| Model size | — | *0.6b* |
| Attention type | — | *Spatio-Temporal Self-Attention / Flash Attention* |
| Training objective | — | Rectified Flow velocity regression |
| Time sampling | — | $t \sim U[0, 1]$ |
| Noise prior | — | $x_0 \sim \mathcal{N}(0, I)$ |
| Batch size (global) | — | *32* |
| Optimizer | — | AdamW |
| Learning rate | $\eta$ | *1e-5* |
| LR schedule | — | *cosine + warmup* |
| Warmup steps | — | *5k* |
| Gradient clipping | — | *1.0* |
| Training precision | — | *Mixed precision (bf16)* |
| Training steps / epochs | — | *100* |
| EMA | — | *decay=0.999* |
| CFG (inference) | — | *Swedish = 6,MRMS=4* |
| (B) 2D VAE (Latent autoencoder) | | |
| **Item** | **Symbol / Option** | **Value** |
| Architecture | — | *VideoAutoencoderKL-2D(Rombach et al., 2022)* |
| Downsampling factor | — | $\times 8$ |
| Latent channels | $C_z$ | *4* |
| Encoder/decoder depth | — | *4* |
| KL weight | $\beta$ | *1e-6* |
| Reconstruction loss | — | *L1/L2(fine)* |
| Batch size | — | *24* |
| Optimizer | — | AdamW |
| Learning rate | $\eta$ | *1e-4* |
| Training epochs | — | *50* |
| Training precision | — | *Mixed precision (bf16)* |

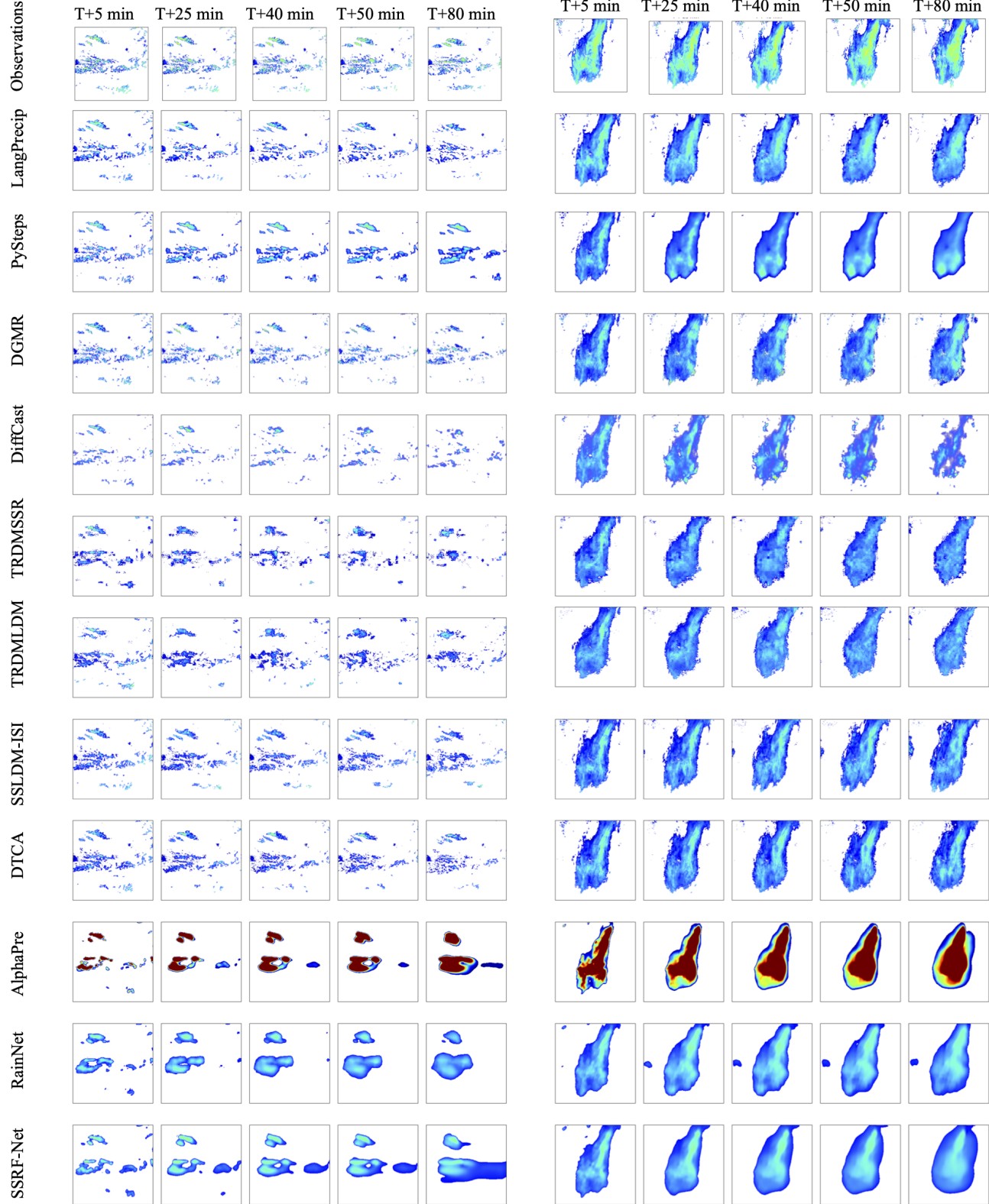

*Figure 19.* Prediction examples on the Swedish dataset. Two representative events are shown, with forecasts at multiple lead times (T+5 to T+80 minutes), illustrating the temporal evolution of precipitation intensity and spatial structure.

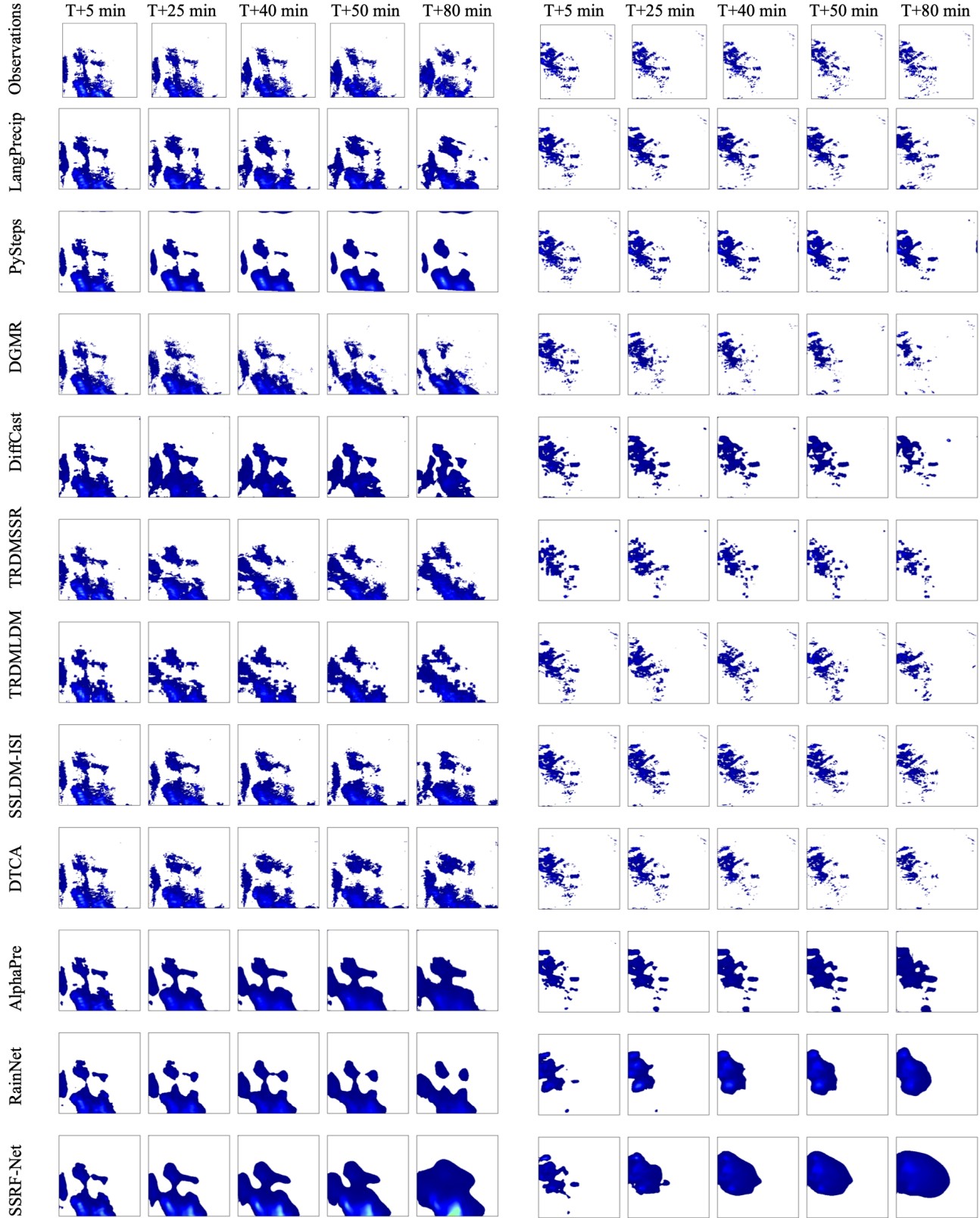

*Figure 20.* Prediction examples on the MRMS dataset. Two representative events are shown, with forecasts at multiple lead times (T+5 to T+80 minutes), illustrating the temporal evolution of precipitation intensity and spatial structure.

## Radar Echo Sequence                    Motion Description

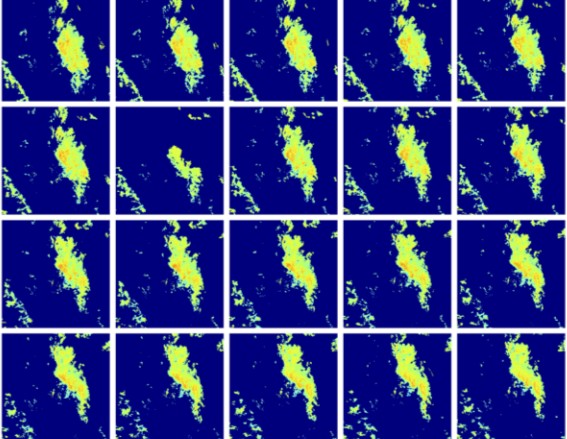

The radar echo forms a vertically elongated, filamentous structure centered in the frame, with a narrow, bright core and diffuse, irregular edges indicating internal substructure, while scattered returns appear along its flanks and upper left. Over the 20-frame sequence, the entire system moves rapidly upward along a stable trajectory near the same direction, showing minimal deviation and high motion coherence Simultaneously, a distinct counterclockwise rotation develops within the central dense region, supported by a strong curl signal, while divergence analysis reveals net contraction, reflecting inward flow and progressive tightening of the structure. As the system ascends, its horizontal extent gradually narrows and the core intensifies, maintaining compactness without lateral spreading or fragmentation. Throughout the sequence, the echo preserves its orientation and integrity, driven by a combination of sustained upward translation, cyclonic rotation, and convergent dynamics, highlighting a coherent, organized feature evolving under a stable kinematic regime.

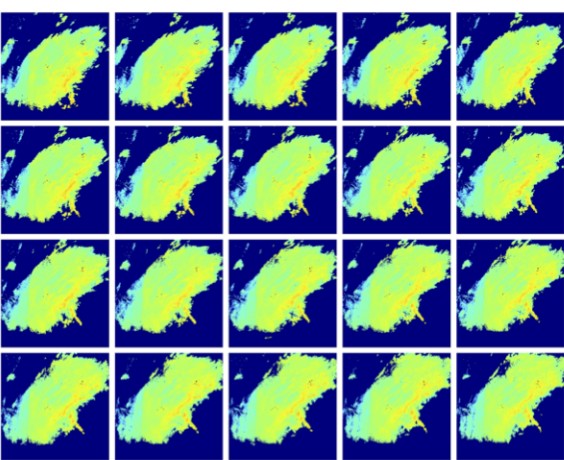

A dominant convective echo clings to the right edge, surging steadily upward and drifting slightly rightward across the frame, its motion smooth and unwavering as if locked on a northeastward trajectory. Bright, dense cores near the top shear into fragmented, fading tendrils that trail downward and leftward, dissolving as they retreat from the advancing mass. Scattered echoes flicker distantly in the upper-left, static and disconnected, never drawn into the main current. As the system glides forward, its edges subtly spiral counterclockwise — a slow, coiling dance visible in the curl of its contours and faint internal swirls — while its body pulses gently, expanding and contracting like breath, never rupturing, never scattering. Rotation hums beneath the surface, moderate but secondary to the relentless upward-rightward glide, each frame carrying it further without falter, its form holding firm against minor ripples of deformation, translating more than twisting, yet never still.

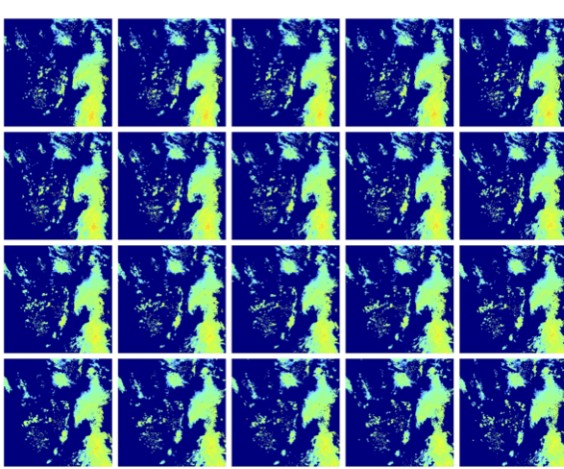

The large, irregularly shaped echo structure drifts steadily toward the upper-right quadrant , with high directional coherence and minimal divergence, indicating strong translational dominance. Its blob-like form, featuring a diffuse, textured interior and uneven, non-linear edges, maintains structural integrity throughout, undergoing only minor morphological adjustments such as localized deformation and slight edge expansion without significant fragmentation or growth. A subtle but measurable counterclockwise rotational component is evident, particularly in the shearing along the trailing edge, suggesting weak vortical dynamics superimposed on the primary motion. Despite internal reorganization remaining limited, gradual tapering of reflectivity from core to periphery and persistent shape evolution reflect a coherent, organized convective system advecting rapidly in a northerly direction, combining robust translation with slight rotational influence and overall stability in form and trajectory.

*Figure 21.* Examples from the LangPrecip-160K dataset, illustrating radar precipitation events and the corresponding motion descriptions.

