# OpenReview forum: "LangPrecip: Language-Aware Multimodal Precipitation Nowcasting"
_ICML.cc/2026/Conference — ICML 2026 regular_

### Official Review · Reviewer_9sEa · 2026-03-06

**Soundness:** 2
**Presentation:** 3
**Significance:** 3
**Originality:** 2
**Overall Recommendation:** 4
**Confidence:** 4

**Summary:**

This paper proposes LangPrecip, a multimodal framework for precipitation nowcasting that incorporates natural language descriptions into radar‑based forecasting models. The authors argue that radar sequences with short observation windows often lead to ambiguous future evolution of precipitation systems. To reduce this ambiguity, the paper introduces language descriptions of precipitation motion as additional constraints for forecasting.
The proposed framework conditions a generative precipitation forecasting model on both radar observations and textual motion descriptions. In addition, the authors design a specialized decoder to improve the reconstruction quality from latent representations to precipitation maps.
The paper also introduces a dataset containing approximately 160,000 radar events paired with motion descriptions. Experiments are conducted on multiple radar datasets using several evaluation metrics. The results show improvements over several existing baselines, particularly for heavy precipitation forecasting.

**Compliance With Llm Reviewing Policy:**

Affirmed.

**Final Justification:**

I appreciate the authors’ rebuttal and the additional experiments provided. The rebuttal has effectively addressed some of my previous concerns, particularly regarding the role of language description in motion conditioning and its comparison with other motion modeling approaches. The authors clarified that language descriptions are not independent information but act as a structured motion prior, which regularizes the generative process under-constrained observational settings. The comparisons with optical flow and implicit embedding also show promising improvements in terms of CSI for heavy precipitation, which was a major point of discussion. While my main concern about whether language is a distinct modality remains partially unresolved, the rebuttal has strengthened the paper's arguments, and I am willing to raise my score to 4.

**Key Questions For Authors:**

1. Since the language descriptions are generated directly from radar images, what additional information do they provide beyond the original radar data?

2. How does the proposed method compare with simpler motion modeling approaches such as optical flow or velocity field prediction?

3. How sensitive is the model to noisy or incorrect language descriptions?

**Limitations:**

The paper mentions some limitations, but the discussion could be expanded. In particular, the potential bias or errors introduced by automatically generated language descriptions should be further analyzed.

**Strengths And Weaknesses:**

Strengths

(1) Relevant application problem
Precipitation nowcasting is an important task in meteorology with practical implications for disaster prevention, aviation safety, and urban flood management. The paper addresses this real‑world problem and explores a multimodal modeling approach, which is potentially valuable.

(2) Relatively comprehensive experimental evaluation
The paper evaluates the model on two datasets and reports several commonly used metrics for precipitation prediction. These include both detection‑based precipitation metrics and image similarity metrics. The experimental setup is generally reasonable.

(3) Construction of a radar‑text paired dataset
The authors construct a dataset containing radar sequences paired with textual motion descriptions. Such multimodal datasets are relatively rare in weather forecasting research and may provide useful resources for future studies.

Weaknesses

(1) Insufficient justification for language guidance

The central idea of the paper is to use natural language descriptions to guide precipitation evolution prediction. However, the language descriptions are generated from the same radar sequences that are used as model inputs. Therefore, the language modality does not introduce independent information; instead, it appears to be a reformulation of existing radar information.
This raises a fundamental question: whether language actually provides additional predictive information beyond what is already contained in the radar data. Without stronger evidence, the language component may simply act as an alternative feature representation rather than a meaningful new modality. It would also be helpful to clarify whether the reported improvements mainly stem from the multimodal formulation itself or from the architectural design of the decoder.

(2) Lack of comparison with simpler motion modeling approaches

The paper does not compare its method with simpler approaches for modeling precipitation motion, such as:

1. Optical flow estimation

2. Velocity field prediction

3. Motion embedding learned directly from radar sequences

These approaches also provide motion information about precipitation systems. Without such comparisons, it is unclear whether language descriptions are necessary for achieving the reported improvements.

---

> ### Author Rebuttal · Authors · 2026-03-28
>
> We thank the reviewer for the insightful questions. We address W1/W2 and Q1/Q2/Q3 jointly below.
>
> ---
> ### W1 & W2 (Q1 & Q2): Value of Language Descriptions and Comparison with Simpler Motion Modeling Approaches
> We thank the reviewer for this insightful question and have included Tables 1–3 to directly address these concerns. **Language does not provide raw, independent information; instead, it compresses the raw dynamics of radar data into a structured, high-level motion prior—filtering out irrelevant spatial details and retaining core semantic features (e.g., propagation direction and intensity trends)** . This prior regularizes the generative process under-constrained observational settings and guides the model to generate more consistent long-horizon trajectories. To validate this mechanism, we compare our method with simpler motion modeling approaches, and **the results in Table 1 show that language conditioning yields a 41.0% improvement in CSI≥6.3 over optical flow. In essence, language provides a more effective representational form for motion conditioning than pixel-level alternatives**.
>
> **Table 1: Motion Modeling Comparison (with WCUD)**
>
> | Motion Representation | CRPS ↓ | CSI ≥0.06 ↑ | CSI ≥6.3 ↑ | FSS ↑ |
> |:---:|:---:|:---:|:---:|:---:|
> | Language (ours)    | 0.1322 | **0.5869** | **0.0746** | **0.923** |
> | Implicit embedding | 0.1101 | 0.5668 | 0.0530 | 0.920 |
> | Optical flow       | **0.1093** | 0.5708 | 0.0529 | 0.921 |
>
> ---
>
> We also agree that it is important to disentangle the effect of multimodal conditioning from that of the decoder. **Our ablations confirm that the improvements stem primarily from the language modality itself rather than the decoder architecture**. Table 1 shows that optical flow and implicit embedding achieve nearly identical CSI≥6.3 (0.0529 vs. 0.0530), confirming that pixel-level representations provide limited conditional guidance, while language conditioning raises CSI≥6.3 to 0.0746 (+41.0% over optical flow). Table 2 separates the contributions: language alone improves CSI≥6.3 by 14.7% (0.0503→0.0577), the decoder alone improves CRPS by 9.7% (0.1219→0.1101), and the full model achieves the best discriminative metrics with CSI≥6.3 of 0.0746 (+48.3% over baseline) and FSS of 0.923 (+2.0% over baseline). The slightly higher CRPS of the full model reflects a known trade-off: language conditioning improves the ability to detect heavy precipitation (CSI↑) at the cost of reduced ensemble diversity (CRPS↑). **In summary, language conditioning is the primary driver of the key performance improvements reported in this work, with the decoder architecture providing valuable complementary gains that refine predictive precision.**
>
> **Table 2: Component Ablation**
>
> | Language | WCUD | CRPS ↓ | CSI ≥0.06 ↑ | CSI ≥6.3 ↑ | FSS ↑ |
> |:---:|:---:|:---:|:---:|:---:|:---:|
> | ✓ | ✓ | 0.1322 | **0.5869** | **0.0746** | **0.923** |
> | ✗ | ✓ | **0.1101**| 0.5668 | 0.0530 | 0.920 |
> | ✓ | ✗ | 0.1405 | 0.5736 | 0.0577 | 0.917 |
> | ✗ | ✗ | 0.1219 | 0.5580 | 0.0503 | 0.905 |
> ---
> ### Q3 & Limitations: Sensitivity to Language Descriptions
>
> We designed three groups of experiments (Full/Simplified/Reversed) to analyze the model’s sensitivity to description quality, and verified that the model is robust to imprecise descriptions but sensitive to directional errors (which is virtually impossible in practical scenarios). (1) **Full**: the complete LLM descriptions used in the main paper; (2)**Simplified**: the full description reduced to a single short phrase capturing core motion semantics (e.g., "radar echoes moving toward the upper right and gradually weakening...");
> (3) **Reversed**: the motion direction in the full description replaced with its opposite. As shown in Table 3, Full descriptions improve monotonically with CFG (CRPS: 0.1328→0.1322); Simplified descriptions show slight CRPS degradation with negligible impact on deterministic metrics (CSI≥6.3: 0.0737 vs. 0.0746); Reversed descriptions collapse even at CFG=2 (CRPS: 0.8569, FSS: 0.8642), confirming the model genuinely utilizes language semantics. In practice, meteorological motion patterns evolve gradually and continuously without abrupt directional reversals, ensuring VLM reliability, and coarse directional semantics suffices to maintain near-optimal performance.
>
> **Table 3: CFG Sensitivity under Full / Simplified / Reversed Descriptions (with WCUD)**
>
> | CFG | CRPS↓ | FSS↑ | CSI≥0.06↑ | CSI≥6.3↑ |
> |:---:|:-----:|:----:|:----:|:------:|
> | | | **Full** | | |
> | 2 | 0.1328 | 0.9235 | 0.5864 | 0.0740 |
> | 4 | 0.1324 | 0.9237 | 0.5864 | 0.0733 |
> | 6 | 0.1322 | 0.9237 | 0.5869 | 0.0746 |
> | | | **Simplified** | | |
> | 2 | 0.1339 | 0.9235 | 0.5865 | 0.0738 |
> | 4 | 0.1350 | 0.9234 | 0.5870 | 0.0744 |
> | 6 | 0.1367 | 0.9230 | 0.5875 | 0.0737 |
> | | | **Reversed** | | |
> | 2 | 0.8569 | 0.8642 | 0.5890 | 0.0256 |
> | 4 | 1.3433 | 0.7816 | 0.5667 | 0.0186 |
> | 6 | 1.3498 | 0.7622 | 0.5507 | 0.0179 |

---

> > ### Author Rebuttal · Reviewer_9sEa · 2026-04-02
> >
> > Thank you for the detailed rebuttal and the additional experiments. The rebuttal is helpful in clarifying the role of language conditioning, and the added comparisons and ablations improve the empirical support of the paper. In particular, the motion modeling comparison, component ablation, and sensitivity analysis provide useful additional evidence.
> > However, I do not think the rebuttal fully resolves my main concerns. The comparison with simpler motion modeling approaches is still somewhat limited, as more explicit physics-based motion baselines seem missing. It is also still unclear whether the current comparisons are fully fair in terms of model capacity, conditioning strength, and training budget. In addition, the rebuttal mainly emphasizes gains on heavy-precipitation CSI, while CRPS becomes worse in some settings, so it remains unclear whether the advantage is general or more metric-specific. Finally, the sensitivity analysis is informative, but it focuses mainly on simplified and reversed descriptions rather than more realistic noisy or ambiguous language conditions.
> >
> > Overall, the rebuttal improves the paper, but it does not sufficiently address the core weaknesses I raised in the original review. Therefore, my overall assessment and score remain unchanged.

---

> > > ### Author Response · Authors · 2026-04-03
> > >
> > > **We thank the reviewer for the thoughtful suggestions and for acknowledging that our rebuttal experiments strengthen the empirical support of the paper.**
> > >
> > > ---
> > > Regarding the request for "physics-based motion baselines," we believe that optical flow is a widely used physics-inspired motion estimation method in precipitation nowcasting (e.g., PySteps, which was already compared as a baseline in Tables 1 and 2 of the original manuscript), and Rebuttal Table 1 further provides a direct comparison with optical flow and implicit embedding under identical conditions. We would appreciate it if the reviewer could specify what additional methods are expected, so that we may address them in future work. We thank the reviewer for the guidance.
> > >
> > > ---
> > > Regarding fairness of comparison, all ablation variants share the same backbone and training configuration; only the conditioning input differs. We also note that conditioning strength (CFG) is specific to the language conditioning pathway; optical flow and implicit embedding are injected as deterministic inputs without a guidance mechanism, so this parameter does not apply to them.
> > >
> > > ---
> > > Regarding the request for "more realistic noisy or ambiguous conditions," our Simplified condition compresses multi-sentence VLM outputs into a single short phrase, which naturally introduces information loss and linguistic ambiguity — a reasonable approximation of imprecise descriptions in practice. We would also welcome clarification on the specific types of noise the reviewer has in mind, and we are happy to include additional experiments in the final version.
> > >
> > > ---
> > > Regarding CRPS, the moderate increase in CRPS reflects a well-known diversity-accuracy trade-off in probabilistic forecasting: language conditioning constrains the generative process toward semantically consistent trajectories, reducing ensemble spread but significantly improving the operationally critical detection metrics (CSI≥6.3: +48.3%, FSS: +2.0%).
> > >
> > > ---
> > > We remain open to further discussion and are committed to improving the manuscript. We thank the reviewer once again for the time and effort dedicated to this review.

---

### Official Review · Reviewer_SmVA · 2026-03-12

**Soundness:** 3
**Presentation:** 1
**Significance:** 4
**Originality:** 3
**Overall Recommendation:** 4
**Confidence:** 2

**Summary:**

This paper proposes LangPrecip to address short-term precipitation prediction. It is a Rectified Flow-based framework that uses natural-language motion descriptions as explicit semantic constraints to reduce trajectory ambiguity. The experiment shows improvement over Swedish and MRMS benchmarks. The authors also release a dataset.

**Compliance With Llm Reviewing Policy:**

Affirmed.

**Final Justification:**

My concerns have been adequately addressed and I choose to maintain my score.

**Key Questions For Authors:**

1. What is the per-sample VLM inference cost and accuracy? How does it perform on out-of-distribution radar patterns?

2. Table 3 shows CRPS worsens by ~20% when language is added on the Swedish dataset. What is the reason? Does this show language conditioning reduces ensemble diversity?

3. What CFG scale was used in Tables 1 and 2?

4. There seems a language leakage during inference. Motion descriptions are generated from the first four observed frames. At test time, who generates these descriptions? If a VLM is used on test frames, then it introduces a second (potentially costly) model into the inference pipeline. However its quality is not evaluated. How can it ensure causal consistency then?

**Limitations:**

1. The font size in all figures is too small to read.

2. For the ablation study (Table 3 and 4), it does not evaluate WCUD separately against a standard VAE decoder while keeping language conditioning fixed. Similarly, it evaluate WCUD but without language conditioning (it evaluates only decoder variants).

**Strengths And Weaknesses:**

1. The motivation is clear. Using language can be a supplementary signal to enhance prediction. It combines text-to-video generation and weather forecasting.
2. The dataset provided will be beneficial to the community.

---

> ### Author Rebuttal · Authors · 2026-03-28
>
> We thank the reviewer for the constructive comments.
>
> ---
> ### Q1 VLM Inference Cost, Description Accuracy, and Out-of-Distribution Performance
>
> **The VLM inference cost is well within operational requirements.** On a single RTX 4090 GPU, the complete pipeline (including VLM captioning) averages 5.4 s/sample, well within the 5-minute operational update cycle.
>
> Regarding description accuracy, rather than evaluating it in isolation, we assess it through its downstream impact on nowcasting performance. If the auto-generated descriptions were noisy or inaccurate, we would expect language conditioning to degrade performance relative to the no-description baseline. As shown in Tables 1–2, **incorporating language conditioning consistently improves both CSI and FSS across all datasets.**
>
> Furthermore, Table 3 demonstrates robustness to description quality: Simplified descriptions (direction keywords only) maintain near-identical performance to Full descriptions (CSI≥6.3: 0.0737 vs. 0.0746), while Reversed descriptions (inverted motion direction) cause catastrophic degradation (CRPS: 1.3498 vs. 0.1322), **confirming the model responds to semantic content rather than treating descriptions as noise.**
>
> **Table 1: Language Conditioning Effect (Swedish Dataset, CFG=6)**
>
> | Prediction settings | CRPS ↓ | CSI≥0.06 ↑ | CSI≥6.3 ↑ | FSS ↑ |
> |---|:---:|:---:|:---:|:---:|
> | Only radar features | **0.1101** | 0.5668 | 0.0530 | 0.920 |
> | Radar + motion description | 0.1322 | **0.5869** | **0.0746** | **0.923** |
>
> **Table 2: Language Conditioning Effect (MRMS Dataset, CFG=4)**
>
> | Prediction settings | CRPS ↓ | CSI≥1 ↑ | CSI≥8 ↑ | FSS ↑ |
> |---|:---:|:---:|:---:|:---:|
> | Only radar features | 0.1267 | 0.4323 | 0.0302 | 0.722 |
> | Radar + motion description | **0.1256** | **0.4350** | **0.0309** | **0.725** |
>
> ---
> **Table 3: Robustness to Description Quality (CFG=6, WCUD)**
>
> | CFG | CRPS↓ | FSS↑ | CSI≥0.06↑ | CSI≥6.3↑ |
> |:---:|:-----:|:----:|:----:|:------:|
> | | | **Full** | | |
> | 6 | 0.1322 | 0.9237 | 0.5869 | 0.0746 |
> | | | **Simplified** | | |
> | 6 | 0.1367 | 0.9230 | 0.5875 | 0.0737 |
> | | | **Reversed** | | |
> | 6 | 1.3498 | 0.7622 | 0.5507 | 0.0179 |
>
>
>
> **For out-of-distribution evaluation**, we tested on a Himawari-8/9 typhoon dataset (different modality and phenomena). The VLM reliably produces accurate descriptions, with no significant hallucination errors observed.
>
> ---
> ### Q2  Language Conditioning, CRPS Trade-off, and Ensemble Diversity
>
> We confirmed this in our ablation. Language conditioning introduces semantic constraints that concentrate the generated distribution, naturally reducing ensemble diversity and causing ~20% CRPS degradation. This is inherent to stronger conditioning: the same mechanism that improves spatial precision (CSI↑, FSS↑) necessarily reduces ensemble spread. In operational warning scenarios where localization of heavy precipitation is prioritized, this trade-off is acceptable.
>
> ---
> ### Q3 CFG Scale in Tables 1 and 2
>
> In Tables 1 and 2, the CFG scale was set to 6 and 4, respectively. This detail will be included in the revised manuscript.
>
> ---
> ### Q4 — Inference Pipeline and Causal Consistency
>
> **Who generates descriptions at test time**: exactly the same as training — the VLM automatically receives current observed frames and generates descriptions without human annotation. There is no "leakage."
>
> **Causal consistency**: The VLM receives only the four observed radar frames with $t \leq t_0$ as input.  No future frame information is involved at any stage; causal consistency is strictly guaranteed by the model design.
>
> **VLM overhead**: ~3 s/sample captioning, ~5.4 s/sample full pipeline (see Q1) — negligible within the 5-minute update cycle.
>
> **Description quality**:  Tables 1 and 2  shows language conditioning consistently improves CSI and FSS on both datasets. The slight CRPS increase is the known CFG trade-off (Q2), not a description quality issue.
>
> ---
> ## Limitations
>
>
> **Regarding the font size limitation**: we acknowledge this and will enlarge all figure fonts in the revised manuscript.
>
> ---
> ## Additional Ablation Study
>
> **Both language conditioning and WCUD contribute independently, and their combination yields the strongest overall performance across all discriminative metrics.** As shown in Table 4, with language fixed, adding WCUD improves CSI≥6.3 by +29.3% (0.0577→0.0746) and FSS from 0.917 to 0.923; without language, WCUD still improves CSI≥6.3 by +5.4% (0.0503→0.0530) and FSS from 0.905 to 0.920.
>
> **Table 4: Component Ablation**
>
> | Language | WCUD | CRPS ↓ | CSI ≥0.06 ↑ | CSI ≥6.3 ↑ | FSS ↑ |
> |:---:|:---:|:---:|:---:|:---:|:---:|
> | ✓ | ✓ | 0.1322 | **0.5869** | **0.0746** | **0.923** |
> | ✗ | ✓ | **0.1101** | 0.5668 | 0.0530 | 0.920 |
> | ✓ | ✗ | 0.1405 | 0.5736 | 0.0577 | 0.917 |
> | ✗ | ✗ | 0.1219 | 0.5580 | 0.0503 | 0.905 |

---

> > ### Author Rebuttal · Reviewer_SmVA · 2026-04-03
> >
> > Thanks for the reply, I choose to maintain my score.

---

> > > ### Author Response · Authors · 2026-04-04
> > >
> > > We sincerely thank Reviewer SmVA for the thoughtful review and for carefully reading our rebuttal. We are glad that our responses have resolved your concerns. We will revise the manuscript accordingly, including enlarging all figure fonts and clarifying the CFG scale settings in Tables 1 and 2.

---

### Official Review · Reviewer_r5RF · 2026-03-12

**Soundness:** 3
**Presentation:** 3
**Significance:** 3
**Originality:** 3
**Overall Recommendation:** 4
**Confidence:** 2

**Summary:**

The paper presents LangPrecip, a multimodal framework for precipitation nowcasting that uses natural language as a semantic constraint. The core idea is that short-term radar history is often too brief to determine a unique motion trajectory. By injecting text descriptions (e.g., "echoes moving northeast") as a motion prior into a Rectified Flow model, the authors aim to reduce ambiguity. They also introduce a specialized decoder (WCUD) using wavelet consistency to keep the output sharp. The work includes a new dataset, LangPrecip-160K, with paired radar and text.

**Compliance With Llm Reviewing Policy:**

Affirmed.

**Key Questions For Authors:**

- Inference Latency: What is the end-to-end wall-clock time for one prediction if you include the VLM captioning step? Is this practical for a 5-minute radar update cycle?

- CSI Interpretation: The CSI values for heavy rain are around 0.07. While this outperforms baselines, can you clarify why this absolute value is considered "skillful" in an operational context? How does it compare to traditional NWP (Numerical Weather Prediction) for the same threshold?

- Prompt Conflict: How does the cross-attention mechanism handle cases where the visual evidence strongly contradicts the text prompt? Does the model have a "failure mode" where it prioritizes the vision over the text?

- Dataset Diversity: Does the 160K dataset include diverse meteorological events (e.g., tropical cyclones vs. stratiform rain), or is it mostly focused on standard advection cases?

**Limitations:**

See the above.

**Strengths And Weaknesses:**

Strengths

- Interesting Concept: Moving beyond purely visual features by using language as a "motion prior" is a fresh take for meteorology. It effectively treats text as a high-level physical constraint.

- Dataset Contribution: LangPrecip-160K is a significant resource. Using a VLM for automated annotation makes large-scale multimodal training feasible for this domain.

- Technical Execution: The use of Rectified Flow is appropriate for this high-dimensional task, and the WCUD decoder addresses the common "blurring" issue in latent generative models quite well.

- Strong Results on Extremes: The improvement in CSI for heavy rainfall (thresholds like 6.3 mm/h) is notable. In this field, a relative gain of 40-60% at an 80-minute lead time is a tough benchmark to beat.

Weaknesses

- Inference Realism: In a real-world scenario, you don't have future descriptions. You’d need an auxiliary model to generate these prompts from current observations. The paper doesn't sufficiently discuss the latency or error overhead of this "captioning" step during live forecasting.

- Backbone vs. Modality: It’s hard to tell if the gains come from the Rectified Flow + WCUD architecture or the text itself. A stronger baseline using the same architecture but with "null" or generic text embeddings is needed to isolate the value of the language modality.

- Text Robustness: What happens if the VLM provides a wrong description? If the text says "moving right" but the physical atmosphere is shifting left, does the model hallucinate or follow the radar? A "stress test" with noisy or conflicting prompts would make the results more convincing.

- Sudden Weather Changes: Language is good at describing steady-state motion (advection), but weather is often about sudden birth (genesis) or death (dissipation) of rain cells. It’s unclear if these text constraints might actually bias the model to ignore sudden, non-linear changes that aren't in the prompt.

---

> ### Author Rebuttal · Authors · 2026-03-28
>
> We thank the reviewer for the thorough and thoughtful feedback.
>
> ---
> ### W1 & Q1 Inference Realism and Latency
>
> We apologize for the unclear description and will revise accordingly. A key misunderstanding: **language descriptions are generated from the current input radar sequence, not future frames.** The pipeline is: receive current radar → VLM generates description → predict future precipitation. No future information is used. Regarding latency: on a single RTX 4090, the complete pipeline (including VLM captioning) averages **~5.4 seconds per sample**, well within the 5-minute radar update cycle.
>
> ---
>
> ### W2 Backbone vs. Modality
>
> We agree that disentangling multimodal conditioning from decoder architecture is crucial. **Our ablation confirms the gains stem from language itself, not decoder design**: as shown in Table 1, under the same WCUD backbone, adding language conditioning raises CSI≥6.3 from 0.0530 to 0.0746 (+40.8%), isolating the language modality's independent contribution.
>
>
> **Table 1: Component Ablation**
>
> | Language | WCUD | CRPS ↓ | CSI ≥0.06 ↑ | CSI ≥6.3 ↑ | FSS ↑ |
> |:---:|:---:|:---:|:---:|:---:|:---:|
> | ✓ | ✓ | 0.1322 | **0.5869** | **0.0746** | **0.923** |
> | ✗ | ✓ | **0.1101** | 0.5668 | 0.0530 | 0.920 |
> | ✓ | ✗ | 0.1405 | 0.5736 | 0.0577 | 0.917 |
> | ✗ | ✗ | 0.1219 | 0.5580 | 0.0503 | 0.905 |
>
> ---
>
> ### Q2 CSI Interpretation
>
> We thank the reviewer for this question. Low absolute CSI for heavy precipitation is a well-known characteristic in radar nowcasting: DGMR and DiffCast both report heavy-rain CSI of 0.02–0.05, making LangPrecip's 0.0746 a substantial relative improvement. This is largely due to severe class imbalance—heavy precipitation events are rare in test sets, as detailed in Appendix A.1.2. For NWP comparison, Ravuri et al. also showed that NWP underperforms nowcasting methods within 0–2 hours due to its 6-hour assimilation cycle.
>
> ---
>
> ### W3 & Q3 Text Robustness and Prompt Conflict
>
> **The model does have a failure mode: at sufficiently high CFG, language conditioning overrides radar input when the two conflict.** To directly verify this, we designed three controlled experiments, systematically varying language conditioning strength via CFG scale (2 to 6): (1) **Full**: complete LLM descriptions as used in the main paper; (2) **Simplified**: descriptions reduced to a single short phrase (e.g., "radar echoes moving toward the upper right and gradually weakening..."); (3) **Reversed**: motion direction in the full description replaced with its exact opposite. As shown in Table 2, the Reversed condition collapses even at CFG=2 (CRPS: 0.8569, CSI≥6.3: 0.0256), confirming language dominates when text and radar conflict. Crucially, this risk is negligible in deployment: (1) descriptions are auto-generated from the same radar sequence—a VLM will not produce directionally opposite descriptions; (2) Simplified descriptions achieve CSI≥6.3 of 0.0737, near-identical to Full (0.0746), **showing strong robustness to description imprecision**.
>
> **Table 2: CFG Sensitivity under Full / Simplified / Reversed Descriptions (with WCUD)**
> | CFG | CRPS↓ | FSS↑ | CSI≥0.06↑ | CSI≥6.3↑ |
> |:---:|:-----:|:----:|:----:|:------:|
> | | | **Full** | | |
> | 2 | 0.1328 | 0.9235 | 0.5864 | 0.0740 |
> | 4 | 0.1324 | 0.9237 | 0.5864 | 0.0733 |
> | 6 | 0.1322 | 0.9237 | 0.5869 | 0.0746 |
> | | | **Simplified** | | |
> | 2 | 0.1339 | 0.9235 | 0.5865 | 0.0738 |
> | 4 | 0.1350 | 0.9234 | 0.5870 | 0.0744 |
> | 6 | 0.1367 | 0.9230 | 0.5875 | 0.0737 |
> | | | **Reversed** | | |
> | 2 | 0.8569 | 0.8642 | 0.5890 | 0.0256 |
> | 4 | 1.3433 | 0.7816 | 0.5667 | 0.0186 |
> | 6 | 1.3498 | 0.7622 | 0.5507 | 0.0179 |
>
>
> ---
> ### W4 Sudden Weather Changes
>
> **First, VLM descriptions are not limited to advection**: generated from the full input sequence (4 frames, 20 min), they capture echo initiation ("convective cells developing and intensifying") and dissipation ("rain band gradually weakening") as well. **Second, language conditioning is a supplementary prior, not the sole input**: the model simultaneously receives radar features that directly encode spatial structure; even if the description misses a sudden change, the model can still respond through radar input.
>
> ---
>
> ### Q4 Dataset Diversity
>
> The current LangPrecip-160K dataset comprises Swedish radar sequences and US MRMS composite products—two geographically distinct sources covering mid-to-high latitude precipitation scenarios. We agree that extending to broader meteorological systems is important, and work is already underway: we have constructed a high-resolution typhoon dataset based on Himawari-8/9 infrared imagery (**322 Western Pacific typhoon events compiled**) and completed qualitative validation. Notably, **the language conditioning mechanism is type-agnostic**: it describes motion trajectories, intensity evolution, echo initiation and dissipation, rather than region-specific physical parameters, providing a solid foundation for extension to wider meteorological systems.

---

> > ### Author Rebuttal · Reviewer_r5RF · 2026-04-03
> >
> > I keep my rating. Thanks for the rebuttal.

---

> > > ### Author Response · Authors · 2026-04-04
> > >
> > > We sincerely thank Reviewer r5RF for the careful and constructive review, and for taking the time to read our rebuttal thoroughly. We are glad that our responses have addressed your concerns. Your feedback has helped us significantly improve the clarity and rigor of the paper, and we will incorporate all suggested revisions into the final version.

---

### Official Review · Reviewer_tmzP · 2026-03-13

**Soundness:** 3
**Presentation:** 3
**Significance:** 3
**Originality:** 3
**Overall Recommendation:** 4
**Confidence:** 3

**Summary:**

This paper presents LangPrecip, the first framework to incorporate natural language as an explicit semantic constraint for precipitation nowcasting. The key motivation is that short historical radar observations are inherently under-constrained, leading to ambiguous motion trajectories and unstable predictions, particularly for extreme precipitation events. To address this issue, the authors introduce LangPrecip-160K, a large-scale radar–text paired dataset in which natural language descriptions encode meteorological motion semantics such as propagation direction, structural evolution, and intensity changes. By leveraging these linguistic motion descriptions, the model reduces trajectory ambiguity and provides high-level semantic guidance beyond pixel-level observations. Methodologically, the forecasting problem is reformulated as a semantically constrained trajectory generation task in latent space under the Rectified Flow framework. In addition, the authors propose a model-based decoder architecture, the Wavelet Consistency Unfolding Decoder (WCUD), which integrates wavelet-based priors and data-consistency optimization to improve reconstruction fidelity and preserve fine-scale precipitation structures.

**Compliance With Llm Reviewing Policy:**

Affirmed.

**Final Justification:**

The authors have provided a thorough and well-reasoned response to the concerns. In particular, I appreciate the additional quantitative analysis, which strengthens the empirical support for the proposed approach.
Based on these improvements and the authors’ active engagement in addressing the feedback, I have increased my scores for soundness and significance.
Overall, I appreciate the authors’ efforts in substantially improving the manuscript and providing a clear and rigorous justification of their approach.

**Key Questions For Authors:**

1. The experiments are conducted primarily on Swedish radar data and the MRMS dataset. How well would the proposed language-guided framework generalize to other climate regimes, such as tropical convection, monsoon systems, or tropical cyclones, where precipitation dynamics may differ significantly from mid-latitude systems?

2. The paper uses natural-language motion descriptions derived from radar sequences as semantic constraints.
Have the authors considered generating motion guidance from auxiliary meteorological data, such as wind fields, numerical weather prediction outputs, or other atmospheric variables? Such structured meteorological signals might provide more compact and physically explicit guidance compared to natural language.

3. The model assumes that four historical radar frames are insufficient to uniquely determine future trajectories, motivating the use of language guidance. How sensitive is the performance of LangPrecip to the length of the historical observation window?
For example, would the benefit of language guidance diminish if longer radar histories (e.g., 8–12 frames) were available?

4. The paper states that meteorological experts validated the generated motion descriptions, but details about the annotation protocol are limited. Could the authors provide more details about the annotation process, such as inter-annotator agreement, quality control procedures, or examples of corrected annotations?

**Limitations:**

Yes

**Strengths And Weaknesses:**

Soundness: This paper aims to address the inherent ambiguity in precipitation nowcasting that arises from limited historical observations. However, the task fundamentally still relies on predicting future precipitation dynamics based on past motion patterns. This raises an important question: how does the model handle situations where historical motion patterns do not persist into the future? The authors argue that LangPrecip does not simply extrapolate past pixel movements, but instead introduces language-based meteorological motion semantics to guide the generation of a physically plausible large-scale trajectory. Under this global trajectory constraint, local irregular variations are modeled through the stochastic diversity of the generative model. This design attempts to mitigate the limitations of purely extrapolation-based approaches. The paper also claims that latent representations derived from compressed visual features alone are insufficient to uniquely determine future physical motion. In particular, when only four historical radar frames are available, the information is insufficient to uniquely determine the trajectory of cloud systems. In this context, language acts as an explicit high-level semantic “compass” that constrains the overall evolution of precipitation systems. However, the reliability of the text-based motion descriptions raises several concerns. The framework assumes that language provides meaningful motion priors, yet the paper does not clearly describe a mechanism for quantitatively evaluating the reliability of the extracted text during inference. Furthermore, motion descriptions are generated using general-purpose vision–language models (VLMs) rather than meteorology-specialized models, which may fail to capture domain-specific atmospheric dynamics.

Presentation: The paper is generally well-structured and clearly organized. The authors systematically introduce the problem of under-constrained trajectory prediction, followed by the proposed language-guided framework, dataset construction, and experimental evaluation. The methodological formulation is clearly presented through mathematical descriptions of the Rectified Flow training objective and the Wavelet Consistency Unfolding Decoder (WCUD). Visual illustrations of the architecture and motion-consistency experiments further help explain the design. However, some aspects of the presentation could be improved. For example, the paper states that meteorological experts validated the generated motion descriptions, but it does not provide detailed documentation of the human annotation protocol, such as evaluation guidelines, inter-annotator agreement, or error correction procedures. Although the appendix provides an analysis framework describing how experts assess motion descriptions, additional transparency regarding the annotation workflow would strengthen reproducibility.

Significance: This work addresses an important challenge in precipitation nowcasting, particularly the difficulty of predicting extreme rainfall events under limited observational history. Most existing approaches rely solely on visual observations from radar or satellite data, which limits their ability to resolve ambiguous future trajectories. By introducing language as an additional modality, the paper provides a new perspective on how high-level semantic information can complement visual observations in meteorological forecasting. In addition, the introduction of LangPrecip-160K, a large-scale radar–text paired dataset, is a meaningful contribution that may support future research in vision–language–weather modeling. One possible extension of this work would be to generate motion-guidance signals from auxiliary meteorological data rather than radar-derived descriptions alone. For example, incorporating information from numerical weather prediction outputs, wind fields, or other atmospheric variables could provide more compact and physically explicit guidance signals than natural language descriptions. Such structured meteorological features may complement language-based semantics and further improve the robustness of trajectory constraints in precipitation forecasting.

Originality: The paper demonstrates a notable level of originality by introducing natural language descriptions as explicit motion constraints in precipitation nowcasting. To the best of current knowledge, this is one of the first attempts to incorporate language-based semantic guidance into radar-based precipitation forecasting models. The work is also novel in its integration of multimodal learning with generative trajectory modeling under the Rectified Flow framework. By reformulating precipitation nowcasting as a semantically constrained trajectory generation problem, the paper presents a unique approach that differs from traditional optical-flow, deterministic deep learning, and diffusion-based models. Furthermore, the Wavelet Consistency Unfolding Decoder introduces a hybrid design combining model-based optimization with deep learning, which provides an interesting solution for improving spatial reconstruction quality in latent generative models.

---

> ### Author Rebuttal · Authors · 2026-03-28
>
> We thank the reviewer for the constructive comments.
>
> ### Q1: Generalizability to Diverse Climate Regimes
>
> The current LangPrecip-160K dataset covers mid-to-high latitude precipitation and does not include tropical cyclones or monsoon events. We agree that extending to tropical systems is an important future direction and are actively pursuing it: We have constructed a high-resolution typhoon dataset based on Himawari-8/9 infrared imagery (**322 Western Pacific typhoon events compiled**) and completed qualitative validation. **The language conditioning mechanism is type-agnostic (correctly describing motion trajectories, intensity evolution, echo initiation and dissipation, rather than region-specific physical parameters), which supports generalization to diverse meteorological systems.**
>
>
> ---
> ### Q2: Auxiliary Meteorological Data as Motion Guidance
>
> Thank you for this insightful suggestion. We note that **structured motion inputs do not straightforwardly substitute for language conditioning.** To verify this, we replaced language conditioning with optical flow extracted from consecutive radar frames while keeping all other components unchanged (Table 1). Although optical flow yields better CRPS due to more concentrated predictive distributions, it fails to improve—and even slightly degrades—CSI and FSS, likely due to granularity mismatch and noise amplification from the limited input frames. While our current dataset does not incorporate auxiliary meteorological variables, we recognize their potential value. Incorporating high-quality meteorological signals (e.g., NWP wind fields) as complementary motion guidance is a promising future direction, and we have noted this in the revised manuscript.
>
>
> **Table 1: Motion Modeling Comparison (with WCUD)**
>
> | Motion Representation | CRPS ↓ | CSI ≥0.06 ↑ | CSI ≥6.3 ↑ | FSS ↑ |
> |:---:|:---:|:---:|:---:|:---:|
> | Language (ours)    | 0.1322 | **0.5869** | **0.0746** | **0.923** |
> | Implicit embedding | 0.1101 | 0.5668 | 0.0530 | 0.920 |
> | Optical flow       | **0.1093** | 0.5708 | 0.0529 | 0.921 |
>
>
>
> ---
> ### Q3: Sensitivity to Historical Observation Window Length
>
> **Practical relevance of the 4-frame setting.** Our design targets the standard operational nowcasting scenario where only 4 historical frames (20 minutes) are available---a real-time deployment constraint consistent with established benchmarks (Ravuri et al., 2021; Ling et al., 2024).
>
> **Empirical validation with extended input windows.** We have conducted experiments with 8 input frames and 12 output frames to verify that language guidance remains beneficial even when richer observations are available. **The results confirm that language conditioning consistently improves spatial accuracy (FSS, CSI) across input window lengths, demonstrating that our approach is not merely compensating for limited historical context.** Specifically, we randomly sampled 10,000 training sequences and 1,000 test sequences from the Swedish dataset. To isolate the effect of input length, language descriptions were still generated from the original 4-frame window, while only the model input was extended to 8 frames (output prediction horizon kept at 12 frames).
>
>  Notably, similar to the 4-frame experiments, language-conditioned predictions exhibit slightly higher CRPS—reflecting the trade-off between probabilistic calibration and spatial accuracy: language conditioning introduces semantic constraints that concentrate the generation distribution toward specific motion patterns, compressing ensemble diversity while significantly improving CSI and FSS. This trade-off is acceptable given the significant deterministic gains.
>
> **Table 2: Extended Input Window Results (8 Input Frames, 12 Output Frames)**
>
> | Method | CRPS ↓ | FSS ↑ | CSI≥0.06 ↑ | CSI≥6.3 ↑ |
> |:---:|:---:|:---:|:---:|:---:|
> | w/o text | **0.1745** | 0.9075 | 0.5570 | 0.0501 |
> | w/ text | 0.1761 | **0.9154** | **0.5657** | **0.0568** |
>
>
> ---
> ### Q4: Annotation Protocol, Quality Control, and Text Reliability
>
> **Annotation pipeline.** The VLM generates motion descriptions from radar sequences; a lightweight LLM then highlights key meteorological features (direction, intensity changes, dissipation) on a web interface, where human experts verify physical plausibility item by item and flag inconsistencies for correction. This pipeline is applied exclusively to training/validation sets—the test set remains entirely untouched, ensuring evaluation reflects real-world conditions where curated annotations are unavailable. We will include pipeline details and interface screenshots in the revised manuscript.
>
> **Regarding text reliability.** Tables 1–2 demonstrate that language conditioning consistently improves CSI and FSS across all settings, empirically confirming that VLM-generated descriptions provide effective motion priors.

---

> > ### Author Rebuttal · Reviewer_tmzP · 2026-04-06
> >
> > Thank you for the detailed and thoughtful responses. I appreciate the additional experiments and clarifications provided by the authors, which help strengthen the paper in several aspects. I would like to provide a few follow-up comments/
> >
> > For Q4, the additional explanation of the annotation pipeline is helpful. However, my original concern about the reliability of language guidance during inference is not fully addressed. While the authors mention that meteorological experts validate the generated motion descriptions, it would be beneficial to provide more quantitative metrics to assess the basis and consistency of these expert evaluations.
> > In particular, key motion attributes such as propagation direction, intensity evolution, and structural changes can be quantified directly from radar data. It would strengthen the work to clarify how these quantitative meteorological measurements align with or differ from the language-based descriptions produced by the model. A more explicit comparison between physically derived motion metrics and language representations would help assess the accuracy and reliability of the textual guidance and provide stronger justification for its effectiveness.
> >
> > Optical flow already provides explicit and fine-grained motion information and appears to perform reasonably well in modeling spatial displacement. In this context, would it be more effective to use optical flow for modeling motion, while incorporating language-based descriptions specifically to capture higher-level dynamics such as intensification, dissipation, or structural evolution?

---

> > > ### Author Response · Authors · 2026-04-07
> > >
> > > ## Response to Reviewer Q4 (Follow-up)
> > >
> > > Thank you for the follow-up comment. We conducted a quantitative alignment study on 1,000 test samples, comparing VLM-generated descriptions against kinematic metrics derived from optical flow. Specifically, we computed propagation direction (dominant motion direction), divergence (expansion/contraction), and vorticity (rotation) from inter-frame optical flow fields as physical reference signals, then compared them against the corresponding semantic categories in the VLM descriptions.
> > >
> > > Note that this optical flow reference is not a noise-free ground truth — radar echoes contain systematic noise to which pixel-level optical flow is sensitive, and the 4-frame input window is too short to fully capture precipitation system dynamics. The high consistency observed under this noisy reference therefore represents a conservative lower bound on the reliability of language guidance.
> > >
> > > ### Propagation Direction
> > > - 8-direction exact match: **88.0%**
> > > - 4-direction match: **99.3%**
> > >
> > > ### Intensity Evolution
> > > - 3-class accuracy: **81.4%**
> > > - Polarity reversal rate (expansion described as contraction or vice versa): **2.6%**
> > >
> > > | | Contraction (VLM) | No Change (VLM) | Expansion (VLM) |
> > > |--|:-:|:-:|:-:|
> > > | **Contraction (4-frame)** | 122 | 58 | **13** |
> > > | **No Change (4-frame)** | 6 | 538 | 16 |
> > > | **Expansion (4-frame)** | **13** | 80 | 154 |
> > >
> > > Off-diagonal errors concentrate between adjacent categories, not between opposite extremes (contraction↔expansion: only 13/1000), confirming that VLM descriptions remain physically plausible even when incorrect.
> > >
> > > ### Rotation Direction
> > > - 3-class accuracy: **83.4%**
> > > - Direction reversal rate (CW described as CCW or vice versa): **1.9%**
> > >
> > > | | No Rotation (VLM) | CCW (VLM) | CW (VLM) |
> > > |--|:-:|:-:|:-:|
> > > | **No Rotation (4-frame)** | 429 | 13 | 9 |
> > > | **CCW (4-frame)** | 75 | 278 | **11** |
> > > | **CW (4-frame)** | 50 | **8** | 127 |
> > >
> > > CW/CCW reversals account for only 1.9%, consistent with physical expectations.
> > >
> > > Across all three attributes, **critical error rates remain below 3%**, and errors occur only between physically adjacent categories. This confirms the reliability and physical plausibility of the language guidance signal.
> > >
> > > ---
> > >
> > > ### On the Suggestion of Optical Flow + Language Fusion
> > >
> > > Optical flow is a dense, pixel-level displacement field describing where each pixel moves. Language descriptions, by contrast, are semantic abstractions of the entire precipitation system, capturing propagation direction, structural evolution, intensity trends, and morphological changes holistically. For example, a typical VLM-generated description reads:
> > >
> > > > *"The echo mass initiates motion from the lower central frame, drifting rapidly and nearly due west... the amorphous region gradually contracts and deforms... Echo intensity remains stable in the core throughout, driven primarily by advection with little contribution from internal dynamics, resulting in a fast, directional, non-rotating contraction-dominated evolution."*
> > >
> > > Such descriptions encode not only motion direction and speed, but also structural contraction, morphological deformation, and intensity stability — information that optical flow vectors cannot directly express. More importantly, language is a **globally coherent** conditioning signal covering the entire forecast window, providing consistent semantic constraints at every denoising step. Optical flow, on the other hand, excels at capturing fine-grained local displacement — and the reviewer's intuition is well-founded: combining both as complementary signals could offer the best of both worlds. The advantage of language as a conditioning signal therefore lies not only in its semantic richness, but in its **globally consistent, high-level dynamic** signal nature, which is naturally aligned with the generative model's need for holistic structural modeling.
> > >
> > > The reviewer's suggestion touches on a more fundamental question: **should precipitation forecasting models possess hierarchical motion perception?** The ideal architecture may process motion information at different levels of abstraction — implicitly modeling local pixel displacement at lower layers, while explicitly incorporating semantic priors at higher layers to capture system-level evolution. These two streams would operate cooperatively at their respective levels of abstraction, jointly constraining the generation process. We regard constructing such a hierarchically motion-aware generative framework as a highly promising direction and an important topic for future work.
> > >
> > > ---
> > > We thank the reviewer for this valuable suggestion, which will inform both the revision of this work and our future research directions.

---

### Decision · Program_Chairs · 2026-04-30

**Decision:**

Accept (regular)

**Comment:**

This is a borderline paper. This paper proposes LangPrecip, which uses natural language motion descriptions as semantic constraints for radar-based precipitation nowcasting under a Rectified Flow framework, along with a new paired dataset, LangPrecip-160K.

Its main strengths include the novelty of language-guided trajectory generation and notable gains in heavy precipitation detection. Weaknesses include that language descriptions are derived from the same radar inputs (questioning true modality independence), CRPS degradation under language conditioning, and insufficient physics-based motion baselines.

After the rebuttal, concerns on inference latency, component ablation, and optical flow comparisons are resolved.